# STOCHASTIC TWO POINTS METHOD FOR DEEP MODEL GRADIENT FREE OPTIMIZATION

## ABSTRACT

Large foundation models, such as large language models, have performed exceptionally well in various application scenarios. Building or fully fine-tuning such large models is usually prohibitive due to either hardware budget or lack of access to backpropagation. The zeroth-order methods offer a promising direction for tackling this challenge, where only forward passes are needed to update the model. This paper introduces an efficient Stochastic Two-Point (S2P) approach within the gradient-free regime. We present the theoretical convergence properties of S2P under the general and relaxed smoothness assumptions. The theoretical properties also shed light on a faster and more stable S2P variant, Accelerated S2P (AS2P), through exploiting our new convergence properties that better represent the dynamics of deep models in training. Our comprehensive empirical results show that AS2P is highly effective in optimizing objectives for large deep models, including language models, and outperforms standard methods across various model types and scales, with $2\times$ speed-up in training over most conducted tasks.

## 1 INTRODUCTION

Utilizing pre-trained large models for various downstream tasks has emerged as a prominent trend, particularly in the context of Large Language Models (LLMs), which demand substantial computational resources and data during their initial training phase (Devlin et al., 2018; Bommasani et al., 2021). Different from smaller deep models, full fine-tuning these models is often prohibitive due to the massive computing resources needed. Therefore, techniques such as parameter-efficient tuning, including prompt tuning (Lester et al., 2021) and LoRA (Hu et al., 2021), as well as zeroth-order methods (Malladi et al., 2023; Prasad et al., 2022), are developed and demonstrated satisfactory performance. Among these approaches, zeroth-order methods have become especially attractive recently since they only rely on function values, often referred to as zeroth-order information, to optimize models, avoid memory-intensive back-propagation, and enable full or partial fine-tuning with minimum computing resources. The line of research is broadly investigated and generally analyzed within the framework of optimizing the non-convex optimization problem $\min_{\mathbf{x} \in \mathbb{R}^d} f(\mathbf{x})$, where the $f : \mathbb{R}^d \to \mathbb{R}$ is differentiable and the derivatives are not directly accessible. The complexity of this problem is studied over function query complexity, namely the complexity in terms of the number of function evaluations.

Existing analyses of zeroth-order approaches mainly focus on convergence to $\epsilon$-first-order stationary points under the general smoothness assumption (Nesterov & Spokoiny, 2017; Bergou et al., 2020). Zeroth-order optimization can be categorized into two types by whether or not it explicitly approximates gradient: gradient estimator and direct search (Ghadimi & Lan, 2013; Chen et al., 2020; Lewis et al., 2000; Conn et al., 2009). *Gradient estimator* methods compute an estimate of the gradient through zeroth-order information to optimize $f$, i.e., random (gradient-free) oracles. Random oracles are analyzed in the framework of Stochastic Approximation (SA), e.g., random-directions SA (RDSA). Gaussian smoothing is a gradient estimator algorithm that initially uses RDSA as a random oracle, and their work establishes the framework of analyzing the convergence properties of $f$ once explicitly obtaining mean squared error between the approximated gradient and true gradient (Nesterov & Spokoiny, 2017). On the other hand, the *direct search* generally optimizes $f$ by updating the objective function along fixed or randomized directions with fixed or adaptive step size (e.g., reduce step size when the selected directions get rejected) (Vicente, 2013). Stochastic Three Points (STP) (Bergou et al., 2020) is a representative approach in this category. With the condition

$\mathbb{E}|\mathbf{s}^T \nabla f(\mathbf{x})| \geq C||\nabla f(\mathbf{x})||$ where $\mathbf{s}$ is a random vector sampled from specific distributions and $C$ is a small positive constant, one of the STP directions $\pm\mathbf{s}$ with an appropriate step size consistently decreases the objective function in expectation.

In practice, it is often useful to sample a symmetric (two-sided) random perturbation per update for optimization problems. This approach finds practical utility in scenarios like LLM fine-tuning (Malladi et al., 2023; Zelikman et al., 2023) and provides theoretical enhancement when exploiting multiple random perturbations per update (Salimans et al., 2017; Mania et al., 2018). Examples include STP from direct search and two-sided *Gradient Approximation* Gaussian smoothing (basic random search), abbreviated as GA. When using symmetric perturbations, their respective updates are given by:

$$\text{STP:} \quad \mathbf{x}_{k+1} = \arg\min\{f(\mathbf{x}_k + \alpha\mathbf{s}_k), f(\mathbf{x}_k - \alpha\mathbf{s}_k), f(\mathbf{x}_k)\},$$

$$\text{GA:} \quad \mathbf{x}_{k+1} = \mathbf{x}_k - \alpha\mathbf{g}_k \text{ where } \mathbf{g}_k = \frac{f(\mathbf{x}_k + \rho\mathbf{s}_k) - f(\mathbf{x}_k - \rho\mathbf{s}_k)}{2\rho}\mathbf{s}_k, \mathbf{s}_k \sim \mathcal{N}(0, \mathbf{I}),$$

where $\alpha$ denotes step size and $\rho$ denotes smoothing parameter. Note that $f(\mathbf{x}_k)$ in STP cannot be reused from the previous iteration when using batch data settings. GA and STP have similar convergence and behavior, and we show later in our paper that they can be linked or interconnected under specific settings. The convergence of both approaches relies on the general smoothness assumption, a widely employed concept in non-convex optimization.

In this paper, we advance the efficiency of the zeroth order optimization by proposing a new approach called Stochastic Two-Point (S2P), which eliminates the non-updating component $f(\mathbf{x}_k)$ of STP and thus effectively saving one forward pass in a batch data forward pass. We show that the proposed S2P reaches $\epsilon$-first-order stationary point under both general smoothness assumption and relaxed smoothness assumption ($L_0, L_1$-smoothness (Zhang et al., 2019)). As compared to general smoothness, relaxed smoothness is a more realistic assumption for many real-world tasks, especially for deep models (Zhang et al., 2019; 2020; Danilova et al., 2022). The paper has the following contributions to zeroth-order methods for large deep models:

- We analyze the convergence properties of S2P under general and relaxed smoothness assumptions. The basic form of S2P has query complexity $\mathcal{O}(\frac{d}{\epsilon^2})$ under general smoothness assumption, which is the same with Nesterov & Spokoiny (2017); Bergou et al. (2020).To our knowledge, the analysis of query complexity under the relaxed smoothness assumption is novel.
- Based on our theoretical analysis, we proposed a faster variant, Accelerated S2P (AS2P), which exploits our new convergence properties and incorporates our theoretical findings.
- We conduct extensive experiments on large deep models, including language models, that show AS2P significantly outperforms competing methods on gradient-free adaptation, with $2\times$ speed-up in training over most conducted tasks.

## 2 RELATED WORK

Extensive existing literature studied the zeroth-order optimization under convex and non-convex settings (Shamir, 2017; Jamieson et al., 2012; Agarwal et al., 2009; Raginsky & Rakhlin, 2011; Duchi et al., 2015). Bounds to reach first-order stationary points under general smoothness assumption have been derived, which generally depend on model parameter dimension $d$ (Nesterov & Spokoiny, 2017; Bergou et al., 2020). A line of work investigates the effectiveness of noise perturbation to various tasks, e.g., generalizing Gaussian Smoothing to Bernoulli($\pm1$) distribution (Gao & Sener, 2022), orthonormalization of noise perturbation over Gram–Schmidt process (Choromanski et al., 2018; Maheswaranathan et al., 2019). Moreover, practical and theoretical results showed the advantages of the zeroth-order method meeting low-rank structures of the underlying problem (Cai et al., 2022; Malladi et al., 2023; Wang et al., 2018; Sener & Koltun, 2020). Some approaches also guarantee second-order convergence (Lucchi et al., 2021; Zhang & Gu, 2022; Ren et al., 2023). However, the problem has rarely been studied under the popular relaxed smoothness assumption (Zhang et al., 2019). Based on the theories, many work proposed practical methods to adapt to various deep model scenarios such as hyper-parameter optimization (Bergstra & Bengio, 2012; Yang & Shami, 2020), black-box adversarial attack on deep models (Ilyas et al., 2018; Guo et al., 2019; Liu et al., 2018). Moreover, several methods have been developed for and adapted to deep models gradient-free adaptation (Malladi et al., 2023; Prasad et al., 2022; Deng et al., 2022).

## 3 ACCELERATED STOCHASTIC TWO-POINT SEARCH

In this section, we first introduce a prototype of Stochastic Two-point Search (S2P) and analyze its convergence using the general smoothness assumption. We then improve our analysis of S2P using the relaxed smoothness assumption, which leads to the Accelerated Two-Point Search (AS2P).

Throughout this paper, we use bold lowercase letters $\mathbf{x}, \mathbf{y}$ to denote vectors. For vectors, we use $||\cdot||$ to denote the $\ell_2$-norm. For a function $f : \mathbb{R}^d \to \mathbb{R}$, we use $\nabla f$ to denote the gradient and $f^\star$ to denote the global minimum of function $f$. We use $\mathcal{O}(\cdot), \Omega(\cdot)$ to hide absolute constants that do not depend on any problem parameter. We need the following standard definitions and assumptions (Nesterov & Spokoiny, 2017; Bergou et al., 2020; Zhang et al., 2019).

**Definition 3.1.** For a differentiable function $f$, $\mathbf{x}$ is a $\epsilon$-**first-order stationary point** if $||\nabla f(\mathbf{x})|| \leq \epsilon$.

**Definition 3.2.** A differentiable function $f$ is $L$-**gradient Lipschitz** if $||\nabla f(\mathbf{x}_1) - \nabla f(\mathbf{x}_2)|| \leq L||\mathbf{x}_1 - \mathbf{x}_2|| \quad \forall \mathbf{x}_1, \mathbf{x}_2$.

**Definition 3.3.** A differentiable function $f$ is $(L_0, L_1)$-**smoothness** if $||\nabla^2 f(\mathbf{x})|| \leq L_0 + L_1||\nabla f(\mathbf{x})||$.

**Assumption 1.** *The function $f$ is $L$-gradient Lipschitz.*

**Assumption 2.** *The function $f$ satisfies $(L_0, L_1)$-smoothness*

Unless otherwise specified, we assume function $f$ is bounded below by $f^\star$.

### 3.1 STOCHASTIC TWO-POINT SEARCH (S2P)

We first propose a prototype algorithm, Stochastic Two-Point Search (S2P), which improves STP by removing the non-updating component, $f(\mathbf{x}_k)$. This seemingly minor change eliminates the need for an additional forward pass at each iteration when compared to methods like GA. The change is also *non-trivial* because the computation of $f(\mathbf{x}_k)$ in STP cannot be reused from the previous iteration under the batch data condition, and is critical to the convergence of STP. If a similar convergence is maintained in S2P, such an elimination can greatly reduce the computation needed to optimize large deep models, including language models. The S2P algorithm is summarized in Alg. 1.

Specifically, the choice of the distribution of random perturbations within three commonly used probability distributions, normal, uniform, and Rademacher distribution (Bernoulli $\pm 1$ distribution), does not alter our analysis results within our proof framework. However, we use the random perturbations from the Rademacher distribution for our analysis since STP originally utilizes the normal distribution and uniform distribution. We also note that S2P involves two different symmetric perturbations in each iteration, which are utilized for dynamic step size adaptation. This approach necessitates twice the computational cost in each update compared to GA in practical deployment. Ultimately, our goal is to achieve one symmetric perturbation in each iteration in our proposed accelerated variant of S2P, i.e., AS2P in Alg. 2.

---

**Algorithm 1** Stochastic Two-Point search (S2P).

**Inputs**: Epochs $K$, objective function $f$ parameterized with $\mathbf{x} \in \mathbb{R}^d$, stopping criterion $\epsilon$.
**Parameter**: $\mathbf{x}$

1: **for** $k = 0, ..., K$ **do**
2:     $\mathbf{s}_k \sim \mathcal{R}$ {Rademacher distribution. The normal and uniform distribution also apply.}
3:     Choosing one from Option 1-4: Update $\alpha$
4:         Option 1. $\alpha_k = \alpha_0/\sqrt{Kd}$ {Theorem 3.1}
5:         Option 2. $\alpha_k = \frac{|\gamma_k|}{Ld}$ where $|\gamma_k| = \frac{|f(\mathbf{x}+\rho\mathbf{s}_k)-f(\mathbf{x}-\rho\mathbf{s}_k)|}{2\rho}$ {Theorem 3.1}
6:         Option 3. $\alpha_k = \sqrt{2}/BL_1\sqrt{dK}$ {Theorem 3.2}
7:         Option 4. $\alpha_k = \frac{|\gamma_k|}{(AL_0+\sqrt{2}BL_1|\gamma_k|)d}$ where $|\gamma_k| = \frac{|f(\mathbf{x}+\rho\mathbf{s}_k)-f(\mathbf{x}-\rho\mathbf{s}_k)|}{2\rho}$ {Theorem 3.2}
8:     $\mathbf{x}_{k+1} = \arg\min\{f(\mathbf{x}_k + \alpha_k\mathbf{s}_k), f(\mathbf{x}_k - \alpha_k\mathbf{s}_k)\}$
9: **end for**
10: **return x**

---

## 3.2 S2P UNDER GENERAL SMOOTHNESS ASSUMPTION

We first analyze the convergence properties of $f$ running the proposed S2P algorithm under the general smoothness assumption. Similar to STP, we initiate our analysis from Lemma 3.1, which shows the absolute value of the inner product between gradient $\mathbf{g}$ and random perturbation $\mathbf{s}$ is larger than a positive value in expectation, which forms the foundation of descent. Building upon this foundation, Lemma 3.2 introduces a progressive bound and identifies the optimal step size at each iteration. This optimal step size inspires our algorithm development, particularly Option 2 in Theorem 3.1. The central result in this subsection is Theorem 3.1, which establishes that Alg. 1 can employ both stationary and dynamic step sizes (Option 1 and Option 2, respectively) to reach an $\epsilon$-first-order stationary point with a query complexity of $\mathcal{O}(\frac{d}{\epsilon^2})$.

Especially, the strategy of dynamic step size aims to approximate the optimal step size at each iteration, i.e., approximating $\alpha_k^{opt} = \frac{|\nabla f(\mathbf{x}_k)^T \mathbf{s}_k|}{Ld}$ with $\alpha_k = \frac{|\gamma_k|}{Ld}$ where $|\gamma_k| = \frac{|f(\mathbf{x}+\rho\mathbf{s}_k)-f(\mathbf{x}-\rho\mathbf{s}_k)|}{2\rho}$. Simultaneously, the error $|\delta_k| := |\alpha_k - \alpha_k^{opt}| \leq \frac{\rho}{2}$ is controlled. Please refer to inequality (7) in Appendix B.1 for more details. The above findings underline the fundamental correlation between the step-wise step size $\alpha_k$ and $|\gamma_k|$, specifically, $\alpha_k \propto |\gamma_k|$ with a sufficient small $\rho$. Another crucial observation is the interplay between step size $\alpha_k$ and smoothing parameter $\rho_k$ (Note the transition from $\rho \rightarrow \rho_k$ when adopting specific step-wise strategies). Results presented in Appendix B.1 (proof of Lemma 3.2) and Appendix B.1 (proof of Theorem 3.1) show the requirement of both $\alpha_k$ and $\rho_k$ fall within the range of $(0, \frac{\sqrt{2}||\nabla f(\mathbf{x}_k)||}{Ld}]$ to ensure step-wise progress. This observation hints at a significant connection in magnitude between $\alpha_k$ and $\rho_k$, which inspires the development of Alg. 2.

We want to emphasize that our results also reveal the inherent connection between S2P and GA: S2P for Option 2 has almost the same updating formula with GA, when the sign trick described in Section 3.4 is applied.

**Lemma 3.1.** For all $\mathbf{g} \in \mathbb{R}^d$, and random vector $\mathbf{s} \sim \mathcal{R}$ where $\mathcal{R}$ is the Rademacher distribution, then $\mathbb{E}_{\mathbf{s}\sim\mathcal{R}}|\langle \mathbf{g}, \mathbf{s} \rangle| \geq \frac{1}{\sqrt{2}}||\mathbf{g}||_2$.

The result can be directly derived by applying Khintchine inequality (Khintchine, 1923), and the proof is presented in the appendix A. Please refer to Lemma 3.4 in Bergou et al. (2020) for similar results with normal&unifrom distributions. Note that the random perturbation can be normalized as done in STP, so we have $\mathbb{E}_{\mathbf{s}\sim\mathcal{R}}|\langle \mathbf{g}, \frac{\mathbf{s}}{||\mathbf{s}||} \rangle| = \frac{1}{\sqrt{d}}\mathbb{E}_{\mathbf{s}\sim\mathcal{R}}|\langle \mathbf{g}, \mathbf{s} \rangle| \geq \frac{1}{\sqrt{2d}}||\mathbf{g}||_2$. The formula trick can be easily applied to the following analysis, and the conclusion remains the same.

**Lemma 3.2** (Progressive bound). Suppose objective function $f(\cdot)$ satisfies Assumption 1 and $||\nabla f(\mathbf{x}_k)||_2 \geq \epsilon_g$. If we run algorithm 1 with step size $\alpha = \frac{\sqrt{2}\epsilon_g}{2Ld}$, we have following progressive bound $\mathbb{E}[f(\mathbf{x}_{k+1}) - f(\mathbf{x}_k)|\mathbf{x}_k] \leq -\Omega(\frac{\epsilon_g^2}{Ld})$, where $\mathbb{E}[\cdot|\mathbf{x}_k]$ denotes the conditional expectation w.r.t. $\mathbf{x}_k$.

The proof is presented in the appendix B.

**Theorem 3.1** (Query complexity). Suppose objective function $f(\cdot)$ satisfies Assumption 1. If we run algorithm 1 with step size strategy options 1 or 2, the algorithm returns in expectation an $\epsilon$-first-order stationary point in $\mathcal{O}(\frac{d}{\epsilon^2})$ function evaluations. Specifically,

$$\text{For option 1} \quad K \geq \frac{2d}{\epsilon^2}(\frac{(f(\mathbf{x}_0) - f^\star)}{\alpha_0} + \frac{L\alpha_0}{2})^2, \quad \text{For option 2} \quad K \geq \frac{4Ld(f(\mathbf{x}_0) - f^\star)}{\epsilon^2 - \frac{\rho^2}{2}},$$

where $\alpha_0 > 0$ for Option 1 stationary step size; For Option 2 dynamic step size, scalar $\rho_k \in (0, \frac{\sqrt{2}||\nabla f(\mathbf{x}_k)||}{Ld}]$ for $\rho_k$ in each iteration. Generally, it can be set to a small value, e.g., $\rho = \frac{\sqrt{2}\epsilon}{Ld}$.

The proof is presented in the appendix B.

## 3.3 S2P UNDER RELAXED SMOOTHNESS ASSUMPTION

We now analyze the convergence properties of the proposed S2P algorithm under the relaxed smoothness assumption. The assumption posits that $f$ may behave like a smooth function in certain local regions of the loss landscape, but there can also exist some highly non-smooth regions where the

top eigenvalue of Hessian may be large, necessitating special considerations (Zhang et al., 2019; Kunstner et al., 2023).

Lemma 3.3 provides the progressive bound and the optimal step size at each iteration. We highlight that $\epsilon_g$ is no longer linearly dependent on the step size $\alpha$, which distinguishes this result from the one Lemma 3.2. Intuitively, a large $\epsilon_g$ indicates a large gradient, which may, in turn, imply a large top eigenvalue of Hessian under Assumption 2. Consequently, a large step size is no longer the best choice. This concept is pivotal in further improvements upon S2P.

The main result in this subsection is Theorem 3.2, which shows that Alg. 1 can employ both stationary and dynamic step sizes (Option 3 and Option 4, respectively) to reach an $\epsilon$-first-order stationary point with a query complexity of $\mathcal{O}(\frac{d}{\epsilon^2})$. Importantly, Theorem 3.2 shows the structured nature within the learning process when taking dynamic step size. For instance, in regions where the function is smooth and the gradient norm is large, we can anticipate a reduced query complexity. Conversely, under the fourth condition outlined in Table 1, we encounter situations where it is impossible to decrease $||\nabla f(\mathbf{x})||$ due to high levels of non-smoothness. Fortunately, our proposed step size strategy allows us to safely traverse these highly non-smooth regions.

Importantly, theorem 3.3 shows that the gradient norm $||\nabla f(\mathbf{x})||$ is bounded by $|\gamma|$ in expectation. This implies that the statistical information of $\gamma$ is highly likely to reveal characteristics of gradient norm information, which is further correlated with second-order information under Assumption 2. To illustrate, let us define $\tau_k := \eta\sigma_\gamma := \eta \text{Std Dev}(\gamma_{\text{recent}})$, i.e., $\tau_k$ at $k$-th iteration represents $\eta\times$ standard deviation of recent observations (e.g., the most recent 10% iterations) of $\gamma$. Then, with sufficient small $\rho$, Theorem 3.3 suggests that $||\nabla f(\mathbf{x}_k)|| \leq \eta_a\text{Std Dev}(\gamma_{\text{recent}})$ almost for sure with appropriate choice of $\eta_a$, such as $3\sqrt{2}$. Moreover, under Assumption 2, it can establish that $||\nabla^2 f(\mathbf{x})||$ is bounded by $\eta_b\text{Std Dev}(\gamma_{\text{recent}})$ under certain confidence interval with careful selection of $\eta_b$.

**Lemma 3.3** (Progressive bound). Suppose objective function $f(\cdot)$ satisfies Assumption 2 and $||\nabla f(\mathbf{x}_k)|| \geq \epsilon_g$. Alg. 1 with step size $\alpha = \frac{\sqrt{2}\epsilon_g}{2(AL_0+BL_1\epsilon_g)d}$ gives the following following progressive bound $\mathbb{E}[f(\mathbf{x}_{k+1}) - f(\mathbf{x}_k)|\mathbf{x}_k] \leq -\Omega(\frac{\epsilon_g^2}{(AL_0+BL_1\epsilon_g)d})$, where $\mathbb{E}[\cdot|\mathbf{x}_k]$ denotes the conditional expectation w.r.t. $\mathbf{x}_k$, and constants $A = 1.01, B = 1.01$.

The proof is presented in the appendix C.1.

**Theorem 3.2** (Query complexity). Suppose objective function $f(\cdot)$ satisfies Assumption 2. With step size strategy options 3 or 4, Alg. 1 returns in expectation an $\epsilon$-first-order stationary point in $\mathcal{O}(\frac{d}{\epsilon^2})$ function evaluations. Specifically,

$$\text{For option 3} \quad K \geq (\sqrt{d} + \frac{AL_0\sqrt{d} + BL_1(f(\mathbf{x}_0) - f^\star)\sqrt{d}}{\epsilon})^2$$

$$\text{For option 4} \quad \text{The result is summarized in Table 1.}$$

where constants $A = 1.01, B = 1.01$.

The proof is presented in the appendix C.2.

| Conditions[b] | requirement over $\rho$[a] | Query complexity |
|---|---|---|
| $L_1 \leq \frac{1}{\sqrt{2}B}, ||\nabla f(\mathbf{x})|| \geq \frac{AL_0}{1-\sqrt{2}BL_1}$ | $\rho \leq \frac{1}{d\sqrt{2\xi\sqrt{d}}}$ | $\frac{8d(f(\mathbf{x}_0)-f^\star)}{\epsilon}$ |
| $L_1 \leq \frac{1}{\sqrt{2}B}, ||\nabla f(\mathbf{x})|| \leq \frac{AL_0}{1-\sqrt{2}BL_1}$ | $\rho \leq \frac{1}{d}\sqrt{\frac{\epsilon}{2\xi(AL_0+\sqrt{2}BL_1\epsilon)\sqrt{d}}}$ | $\frac{8AL_0d(f(\mathbf{x}_0)-f^\star)}{(1-\sqrt{2}BL_1)\epsilon^2}$ |
| $L_1 \geq \frac{1}{\sqrt{2}B}, ||\nabla f(\mathbf{x})|| \leq \frac{AL_0}{\sqrt{2}BL_1-1}$ | $\rho \leq \frac{1}{d}\sqrt{\frac{\epsilon}{2\xi(AL_0+\sqrt{2}BL_1\epsilon)\sqrt{d}}}$ | $\frac{8AL_0d(f(\mathbf{x}_0)-f^\star)(2\sqrt{2}BL_1-1)}{(\sqrt{2}BL_1-1)\epsilon^2}$ |
| $L_1 \geq \frac{1}{\sqrt{2}B}, ||\nabla f(\mathbf{x})|| \geq \frac{AL_0}{\sqrt{2}BL_1-1}$ | $\rho \leq \frac{1}{d}\sqrt{\frac{\epsilon}{2\xi(AL_0+\sqrt{2}BL_1\epsilon)\sqrt{d}}}$ | $\frac{8(2\sqrt{2}BL_1-1)(\sqrt{2}BL_1-1)(f(\mathbf{x}_0)-f^\star-\epsilon)d}{AL_0}$ |

[a] $\xi$ is a constant associated with third-order property of $f$, detailed in appendix inequality (13).
[b] For forth condition, reaching local $\epsilon$-optimal point instead of $\epsilon$-first-order stationary point, detailed in appendix inequality (17).

Table 1: With dynamic step size strategy, the convergence property of $f$ under relaxed smoothness.

**Theorem 3.3.** Suppose objective function $f(\cdot)$ satisfies Assumption 2. Then the gradient norm $||\nabla f(\mathbf{x})||$ can be bounded in expectation as

$$|\gamma| - \rho d(AL_0 + BL_1||\nabla f(\mathbf{x})||) \leq ||\nabla f(\mathbf{x})|| \leq \sqrt{2}|\gamma| + \sqrt{2}\rho d(AL_0 + BL_1||\nabla f(\mathbf{x})||)$$

where $|\gamma| = \frac{|f(\mathbf{x}+\rho\mathbf{s}) - f(\mathbf{x}+\rho\mathbf{s})|}{2\rho}$. Constants $A = 1.01, B = 1.01$ when $\rho \leq \frac{1}{2L_1 d}$.

The proof is presented in Appendix C.3.

---

**Algorithm 2** Accelerated Stochastic Two-Point search (AS2P).

---

**Inputs**: Epochs $K$, dataset $\mathcal{D}$, objective function $f(\cdot)$ parameterized with $\mathbf{x} \sim \mathbb{R}^d$, scalar $\rho_0$ and $\rho_{\text{end}}$ as smoothing parameters, and scalars $\eta_a, \eta_b$. Decay strategy, e.g., cosine decay.
**Parameter**: $\mathbf{x}$

1: **for** $k = 0, ..., K$ **do**
2: $\quad \mathbf{s}_k \sim \mathcal{R}$ {Rademacher distribution. Normal and uniform distribution also apply.}
3: $\quad \rho_k =$ full $\mathcal{D}$: $\rho_0$, batch $\mathcal{B} \subset \mathcal{D}$: Decay strategy(init $= \rho_0$, end $= \rho_{\text{end}}, k$)
4: $\quad |\gamma_k| = |\frac{f(\mathbf{x}_k+\rho_k\mathbf{s}_k) - f(\mathbf{x}_k-\rho_k\mathbf{s}_k)}{2\rho_k}|, \beta_k = \text{sign}\big(f(\mathbf{x}_k + \rho_k\mathbf{s}_k) - f(\mathbf{x}_k - \rho_k\mathbf{s}_k)\big)$
5: $\quad \sigma_\rho = \text{Std Dev}(\gamma_{\text{recent}})$
6: $\quad \gamma'_k = 1/(1/|\gamma_k| + 1/\tau'_k)$ where $\tau'_k =$ Decay strategy(init $= 2\eta_a$, end $= \eta_a, k$) $* \sigma_\rho$
7: $\quad \alpha_k =$ full $\mathcal{D}$: $\beta_k\rho_k\frac{\gamma'_k}{\tau^b_k}$, batch $\mathcal{B} \subset \mathcal{D}$: $\beta_k*$Decay strategy(init $= \rho_k, k$)$*\frac{\gamma'_k}{\eta_b\sigma_\rho}$
8: $\quad \mathbf{x}_{k+1} = \mathbf{x}_k + \alpha_k\mathbf{s}_k$
9: **end for**
10: **return** $\mathbf{x}$

---

## 3.4 Accelerated Stochastic Two-Point Search (AS2P)

Our convergence analysis of $f$ running S2P under the general and relaxed smoothness assumptions yields insights into a faster variant of S2P, Accelerated S2P (AS2P). AS2P augments stochastic two-point search with dynamic step sizes and incorporates statistical information related to $\gamma_k$ to potentially capture both first-order and second-order dynamics of the objective function $f$. AS2P algorithm is described in Alg. 2 and has two highlighted improvements.

**Progressive $\gamma$-clipping.** The immediate observation stemming from the convergence properties of $f$ running S2P under relaxed smoothness assumption is the non-linear dependence between the approximated optimal step size $\alpha_k$ and $|\gamma_k|$, i.e., $\alpha_k = \frac{|\gamma_k|}{(AL_0 + \sqrt{2}BL_1|\gamma_k|)d} = \frac{1}{(AL_0/|\gamma_k| + \sqrt{2}BL_1)d}$. Specifically, the step size is almost linearly incremental when $|\gamma_k|$ is small, but the increment decreases fast when $|\gamma_k|$ is relatively large. We thus propose a strategy to mimic similar behavior, i.e., $\alpha_k \propto \gamma'_k$ where $\gamma'_k = \frac{1}{1/|\gamma_k| + 1/\tau^a_k}$. $\tau^a_k = \eta_a \text{Std Dev}(\gamma_{\text{recent}})$ practically act as the threshold to estimate the inhibition strength to $|\gamma_k|$. Moreover, inspired by the structure of optimizing $f$ showing by Table 1 along with empirical investigations, we found that $f$ behaves more like satisfying smooth function during the initial stages of training and entering non-smooth regions as training progresses. So, we propose a progressive adjustment of the threshold $\tau^a_k$ over iterations. The complete strategy is elucidated in line-6 of Alg. 2.

**Automatic Learning Rate.** Having $\alpha_k \propto \gamma'_k$, then we analyze the magnitude of step size. From step-wise descent aspect discussed in section 3.2, both $\alpha_k$ and $\rho_k$ are required to be within range $(0, \frac{\sqrt{2}||\nabla f(\mathbf{x}_k)||}{Ld}]$ to guarantee convergence under general smoothness assumption. Meanwhile, under the relaxed smoothness assumption, Theorem 3.2 suggests that $\alpha_k = \mathcal{O}(\frac{1}{d})$, and Theorem 3.3 reveals $\rho_k = \mathcal{O}(\frac{1}{d})$ for a good approximation of gradient norm. So, if we only tune one hyper-parameter, say $\rho_k$ and approximate it well in practice, then a safe criterion for step size is $\alpha_k \leq \rho_k$. Besides that, according to our analysis, the algorithm applying the dynamic step size strategy has the potential to outperform the algorithm with stationary step size. However, the dynamic step size strategy requires twice symmetric perturbations forward passes at each iteration $k$. In order to reduce the number of forward passes, we propose to assign $\beta_k := \text{sign}(f(\mathbf{x} + \rho_k\mathbf{s}_k) - f(\mathbf{x} - \rho_k\mathbf{s}_k))$ as the sign of $f(\mathbf{x} + \alpha_k\mathbf{s}_k) - f(\mathbf{x} - \alpha_k\mathbf{s}_k)$, which we call *sign trick*. Then the calculation of $\arg\min\{f(\mathbf{x} + \alpha_k\mathbf{s}_k), f(\mathbf{x} - \alpha_k\mathbf{s}_k)\}$ is unnecessary since $\mathbf{x} + \beta_k\text{abs}(\alpha_k)\mathbf{s}_k = \arg\min\{f(\mathbf{x} + \alpha_k\mathbf{s}_k), f(\mathbf{x} - \alpha_k\mathbf{s}_k)\}$

where $\text{abs}(\alpha_k)$ is known in practical. However, for a safe sign assignment, principally, it is required $\alpha_k \leq \rho_k$ at least supposing $\rho_k$ is small enough to guarantee the consistency of sign of directional gradient in local regions. Based on the above intuitions, we propose a strategy $\alpha_k = \beta_k \rho_k \gamma_k' / \tau_k^b$ where $\tau_k^b = \eta_b \text{Std Dev}(\gamma_{\text{recent}})$ since $\gamma_k'/\tau_k^b \leq 1$ almost for sure supposing $\eta_b$ is large enough. Meanwhile, the strategy gives $\alpha_k \propto 1/\tau_k^b$, which potentially fits the view of clipping gradient descent, improving the training process by putting constraints on the step size according to the upper bound on the Hessian (Zhang et al., 2019; Kunstner et al., 2023).

Finally, we have $\alpha_k = \frac{\beta_k \rho_k}{\eta_b \sigma_\gamma / |\gamma_k| + \eta_b / \eta_a}$ in short, which emphasizes (a) The non-linear dependence between $\alpha_k$ and $|\gamma_k|$; (b) The interaction between the absolute value and standard deviation of $\gamma_k$.

## 4 EXPERIMENTS

In this section, we conduct an evaluation of the proposed AS2P with commonly used deep models such as ResNet18, ResNet50, ResNet101, ResNet152 (He et al., 2016) and datasets such as CIFAR-10, CIFAR-100 (Krizhevsky & Hinton, 2009). In addition to these evaluations, we also follow the experimental settings described in Malladi et al. (2023) to full fine-tune LLM model OPT-13B (Zhang et al., 2022) on classification tasks {SST-2, RTE} (Socher et al., 2013; Dagan et al., 2005), multiple choice tasks {Copa, ReCoRD} (Roemmele et al., 2011; Zhang et al., 2018), and generation tasks {SQuAD, DROP} (Rajpurkar et al., 2016; Dua et al., 2019). Specifically, we compare the performance of AS2P against several standard methods, including GA (Nesterov & Spokoiny, 2017; Malladi et al., 2023), GA_sign (Gao & Sener, 2022), STP (Bergou et al., 2020), and utilize normal distribution as the default source of random perturbations. For clarity, GA represents the two-side random direction stochastic approximation as the random oracle with tunable hyper-parameters learning rate and smoothing parameter. Notably, GA is principally equivalent to MeZO as presented in Malladi et al. (2023), which reduces memory consumption with implementation trick by doing the twice forward passes sequentially instead of in parallel. It is worth mentioning that this trick is also applied in our implementations and hardware budget refers to Table 4 in Malladi et al. (2023). Furthermore, we introduce AS2P_sign and GA_sign as variants of AS2P and GA, respectively. These variants utilize the random perturbations sampled from the Rademacher distribution. This choice is influenced by some studies that suggest sign variants often exhibit advantages (Gao & Sener, 2022; Kunstner et al., 2023). The details of the setup are summarized in Appendix D.1.

### 4.1 PERFORMANCE COMPARISON WITH STANDARD METHODS

**Performance over common deep models and datasets.** Each row of Figure 1(b) demonstrates the convergence rate under pre-trained ResNet18&CIFAR10, ResNet50&CIFAR10, ResNet101&CIFAR100, and ResNet152&CIFAR100 respectively. Accordingly, each row of Figure 1(a), which is derived from Figure 1(b), demonstrates the training cost ratio (calculating through number of function queries) of reaching specific loss values (epochs) where the ratio {1, 0.8, 0.6, 0.4, 0.2} are costs of GA reaching {500, 400, 300, 200, 100} epochs. Note that STP requires three function queries at each iteration whereas other methods need two, so Figure 1(a) simply counts the ratio of STP as $1.5\times$ original values when deriving from Figure 1(b). We conclude from Figure 1 that the proposed AS2P outperforms all the baselines, which generally requires $0.5\times$ training cost of other methods to reach some specific loss values under most settings. See Figure 5 in Appendix D.2 for additional results under similar settings. We also notice that the performances have no obvious difference between applying random perturbations sampled from normal distribution and random perturbations sampled from Rademacher distribution. However, both outperform the methods applying uniform noise, referring to Figure 4 in Appendix D.2. However, the performance gap between applying different random distributions is not the main focus of this work, so in the following experiment, we only consider random perturbations sampled from normal distribution.

**Performance over fully fine-tuning LLM.** Figure 2(b) and Figure 2(c) show the convergence rate of fully fine-tuning OPT-13B over six language tasks. Figure 2(a) shows the corresponding training cost ratio of reaching specific loss values (epochs) where the ratio {1, 0.75, 0.5, 0.25} are costs of GA with cosine decay LR reaching {20000, 15000, 10000, 5000} mini-batch iterations. Besides, extra baseline GA with a constant learning rate (LR) is added in this experiment, suggested in Malladi et al. (2023). Overall, Figure 2 shows a large performance improvement of the proposed method AS2P

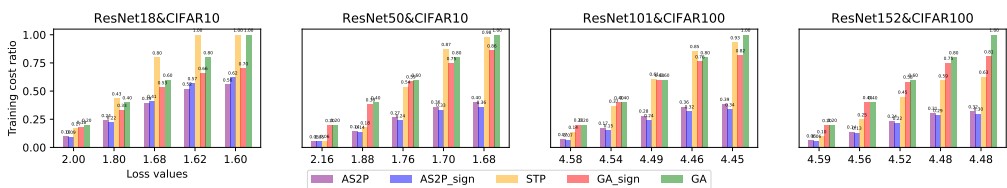

(a) Training cost ratio of reaching specific loss values. Table version in Appendix D.2.

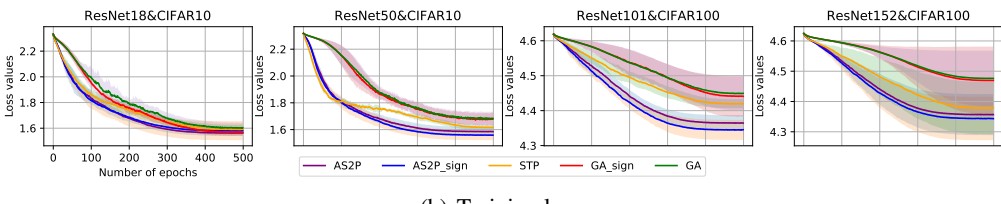

(b) Training loss.

Figure 1: Performance comparison with various baselines under common deep models&datasets.

against other methods. Generally, for most tasks, AS2P requires less than $0.5\times$ training costs of other methods to reach some specific loss values. Specifically, the loss cures between AS2P and STP on task SQuAD largely overlap, however, the actual training cost ratio between AS2P and STP on task SQuAD is around 1:1.5 demonstrated by Figure 2(a).

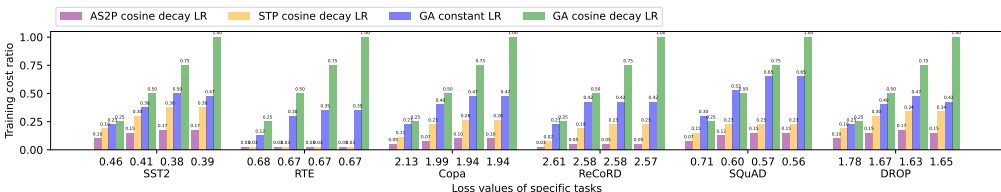

(a) Training cost ratio of reaching specific loss values. Table version in Appendix D.2.

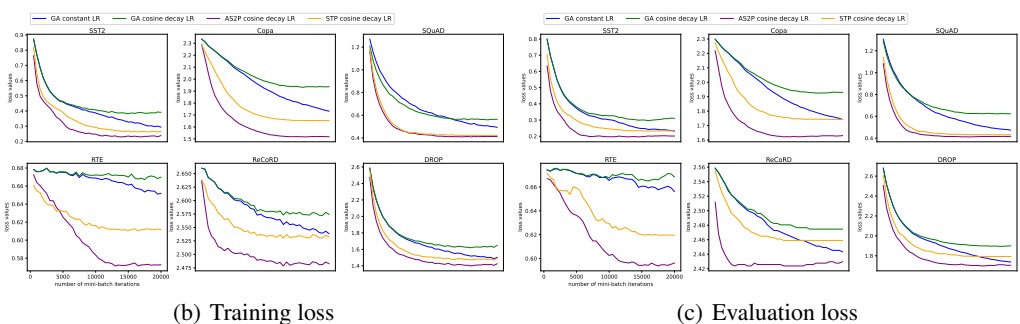

(b) Training loss                                        (c) Evaluation loss

Figure 2: Convergence rate of full fine-tuning OPT-13B model with methods {GA with constant LR}, {GA, STP, and AS2P with cosine decay LR} on classification tasks {SST-2, RTE}, multiple choice tasks {Copa, ReCoRD}, and generation tasks {SQuAD, DROP}.

## 4.2 EFFECTIVENESS OF COMPONENTS IN AS2P

**Automatic learning rate and progressive $\gamma$-clipping.** Further, we verify the effectiveness of two strategies under pre-trained ResNet18 and CIFAR10. Figure 7(a) shows the convergence rate of AS2P without (W.O.) automatic learning rate and AS2P without progressive $\gamma$-clipping. Compared with the GA method, the progressive $\gamma$-clipping strategy, i.e. AS2P W.O. Auto LR, appears to decrease the convergence rate during the initial stages of training but bring a smoother cure. In contrast, the

automatic learning rate strategy, i.e., AS2P W.O. $\gamma$-clipping, increases the convergence rate at the beginning of training. However, it does not show clear advantages in the later stages of training. The proposed AS2P combined the two strategies manages to strike a balance and converges fast throughout the entire training phase, referring to Figure 3(b) for the dynamics of learning rate and $\gamma$.

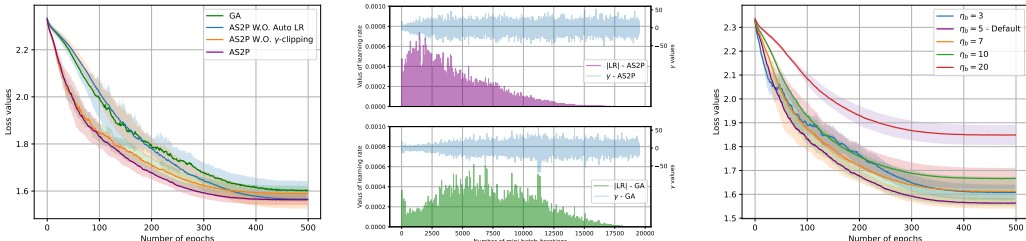

(a) The convergence rate of AS2P without automatic learning rate and without progressive $\gamma$-clipping. (b) The dynamics of learning rate and $\gamma$. (c) Performance comparison with various $\eta_b$.

Figure 3: Verification of effectiveness of components in AS2P

**Hyper-parameters.** Further, AS2P introduces extra hyper-parameters, i.e., $\rho_{\text{end}}$, Std Dev($\gamma_{\text{recent}}$), $\eta_a$ and $\eta_b$. We empirically verified that $\rho_{\text{end}} = \rho_0/10$, $\eta_a = 5$, and utilizing the most recent 10% $\gamma$ for Std Dev($\gamma_{\text{recent}}$) tend to work well across various network architectures and datasets including common deep models and LLMs, as detailed in Table 3 and Table 4 in Appendix D.1. Moreover, the research delves into investigating the impact of the hyper-parameter $\eta_b$ on the convergence rate. Figure 3(c) shows the convergence rate of AS2P applying varying $\eta_b$ under pre-trained ResNet18 and CIFAR10. We note that $\eta_b$ has a relatively significant influence on the convergence rate, which implies the importance of the interaction between the absolute value and standard deviation of $\gamma_k$.

## 5 CONCLUSION AND DISCUSSIONS

In this work, we study the complexity of the proposed S2P method under both general and relaxed smoothness assumptions for gradient-free optimization. Our theoretical analysis induces a variant of S2P, Accelerated S2P, which exploits our new convergence properties and incorporates our theoretical findings, that the standard deviation of $\gamma$ may include second-order information about the objective function $f$. Empirical experiments showed that the proposed AS2P outperforms all baseline methods by a large margin. We note the following important points of discussion.

**Justification for $\eta$Std Dev($\gamma_{\text{recent}}$).** According to studies of clipped gradient descent (Zhang et al., 2019; Kunstner et al., 2023), when crossing the non-smooth regions of loss landscape where the top eigenvalue of Hessian might be large, it is necessary to constrain the step size according to the upper bound of the top eigenvalue of Hessian, the gradient norm. In this context, Theorem 3.3 suggests that $\eta$Std Dev($\gamma_{\text{recent}}$) can serve as a well-estimated upper bound on the top eigenvalue of Hessian. Consequently, it can be used to limit the step size during our symmetric two-point descent, i.e., $\alpha_k \propto \frac{1}{\eta\text{Std Dev}(\gamma_{\text{recent}})}$. What makes this finding intriguing is the phenomenon that it introduces a practical training acceleration. Existing studies suggest using the upper bound on the top eigenvalue of Hessian (the gradient norm) to limit step size to safely traverse non-smooth regions, while naive gradient descent may be too aggressive under the relaxed smoothness assumption. Our work advances this perspective by showing that $\alpha_k \propto \frac{1}{\eta\text{Std Dev}(\gamma_{\text{recent}})}$ can not only impose constraints on the step size along with $\alpha_k \propto \frac{1}{1/\gamma_k+C}$ but also accelerate training process in specific regions. Intuitively, a larger step size is safely expected when the largest Hessian is small.

Further, this work emphasizes the integration between $\alpha_k$, $|\gamma_k|$ and Std Dev($\gamma_{\text{recent}}$). It is non-trivial to consider capturing the interactions through the learning process to avoid tuning hyper-parameters, especially $\eta_b$, for various $f$ or building a more complex relationship instead of such as $\alpha_k \propto \frac{1}{\eta\text{Std Dev}(\gamma_{\text{recent}})}$ only. This investigation will lead us to our future work.

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

## A    TECHNICAL LEMMAS

**Lemma A.1.** ((Zhang et al., 2020) Descent Inequality) Suppose objective function $f(\cdot)$ satisfies Assumption 2, and $c > 0$ be a constant. For any $\mathbf{x}_k$ and $\mathbf{x}_{k+1}$, as long as $||\mathbf{x}_k - \mathbf{x}_{k+1}|| \leq \frac{c}{L_1}$, we have

$$f(\mathbf{x}_{k+1}) \leq f(\mathbf{x}_k) + (\mathbf{x}_{k+1} - \mathbf{x}_k)^T \nabla f(\mathbf{x}_k) + \frac{AL_0 + BL_1 ||\nabla f(\mathbf{x}_k)||}{2} ||\mathbf{x}_{k+1} - \mathbf{x}_k||^2 \quad (1)$$

where $A = 1 + e^c - \frac{e^c - 1}{c}, B = \frac{e^c - 1}{c}$. Note that $A$ and $B$ are monotonically increasing functions w.r.t. $c > 0$.

**Lemma A.2.** (Lemma 3.1) For all $\mathbf{g} \in \mathbb{R}^d$, and random vector $\mathbf{s} \sim \mathcal{R}$ where $\mathcal{R}$ is the Rademacher distribution, i.e., each element $\mathbf{s} \sim \{+1, -1\}$ with equal chances and $\mathbb{E}_{\mathbf{s} \sim \mathcal{R}} ||\mathbf{s}||_2^2 = d$, then $\mathbb{E}_{\mathbf{s} \sim \mathcal{R}} |\langle \mathbf{g}, \mathbf{s} \rangle| \geq \frac{1}{\sqrt{2}} ||\mathbf{g}||_2$.

*Proof.*

$$|\langle \mathbf{g}, \mathbf{s} \rangle| = |\sum_{i=1}^{d} \mathbf{g}_i \mathbf{s}_i| \quad (2)$$

According to Khintchine inequality (Khintchine, 1923), i.e.,

$$A_p (\sum_{i=1}^{d} |\mathbf{g}_i|^2)^{\frac{1}{2}} \leq (\mathbb{E} |\sum_{i=1}^{d} \mathbf{g}_i \mathbf{s}_i|^p)^{\frac{1}{p}} \leq B_p (\sum_{i=1}^{d} |\mathbf{g}_i|^2)^{\frac{1}{2}}$$

where

$$A_p = \begin{cases} 2^{\frac{1}{2} - \frac{1}{p}} & 0 < p < p_0 \\ 2^{\frac{1}{2}} (\Gamma((p+1)/2)/\sqrt{\pi})^{\frac{1}{p}} & p_0 < p < 2 \\ 1 & 2 \leq p < \infty. \end{cases} \quad B_p = \begin{cases} 1 & 0 < p \leq 2 \\ 2^{\frac{1}{2}} (\Gamma((p+1)/2)/\sqrt{\pi})^{\frac{1}{p}} & 2 < p < \infty. \end{cases}$$

where $p_0 \approx 1.847$ and $\Gamma$ is the Gamma function, we have

$$\frac{1}{\sqrt{2}} ||\mathbf{g}||_2 \leq \mathbb{E} |\sum_{i=1}^{d} \mathbf{g}_i \mathbf{s}_i| \leq ||\mathbf{g}||_2,$$

Combined with equation 2, we have

$$\frac{1}{\sqrt{2}} ||\mathbf{g}||_2 \leq \mathbb{E}_{\mathbf{s} \sim \mathcal{R}} |\langle \mathbf{g}, \mathbf{s} \rangle| \leq ||\mathbf{g}||_2.$$

This completes the proof. □

## B    CONVERGENCE ANALYSIS UNDER THE GENERAL SMOOTHNESS ASSUMPTION

### B.1    PROGRESSIVE BOUND OF S2P

**Lemma B.1.** (Lemma 3.2) (Progressive bound) Suppose objective function $f(\cdot)$ satisfies Assumption 1 and $||\nabla f(\mathbf{x}_k)||_2 \geq \epsilon_g$. If we run algorithm 1 with step size $\alpha = \frac{\sqrt{2}\epsilon_g}{2Ld}$, we have following progressive bound $\mathbb{E}[f(\mathbf{x}_{k+1}) - f(\mathbf{x}_k)|\mathbf{x}_k] \leq -\Omega(\frac{\epsilon_g^2}{Ld})$, where $\mathbb{E}[\cdot|\mathbf{x}_k]$ denotes the conditional expectation w.r.t. $\mathbf{x}_k$.

*Proof.* Using $L$-gradient Lipschitz, we have (descent lemma)

$$\mathbb{E}[f(\mathbf{x}_{k+1}) - f(\mathbf{x}_k)|\mathbf{x}_k]$$

$$\leq \mathbb{E}[\nabla f(\mathbf{x}_k)^T(\mathbf{x}_{k+1} - \mathbf{x}_k)|\mathbf{x}_k] + \frac{L}{2}\mathbb{E}[||\mathbf{x}_{k+1} - \mathbf{x}_k||^2]$$

$$= -\alpha\mathbb{E}|\nabla f(\mathbf{x}_k)^T\mathbf{s}_k| + \frac{L\alpha^2}{2}\mathbb{E}||\mathbf{s}_k||_2^2 \quad \text{Take updating step}$$

$$= -\alpha\mathbb{E}|\nabla f(\mathbf{x}_k)^T\mathbf{s}_k| + \frac{L\alpha^2 d}{2}$$

Lemma 2 shows that $\mathbb{E}_{\mathbf{s}_k\sim\mathcal{R}}|\nabla f(\mathbf{x}_k)^T\mathbf{s}_k| \geq \frac{1}{\sqrt{2}}||\nabla f(\mathbf{x}_k)||_2$, then

$$\mathbb{E}[f(\mathbf{x}_{k+1}) - f(\mathbf{x}_k)|\mathbf{x}_k] \leq -\frac{\alpha}{\sqrt{2}}||\nabla f(\mathbf{x}_k)||_2 + \frac{L\alpha^2 d}{2}$$

$$\leq -\frac{\alpha}{\sqrt{2}}\epsilon_g + \frac{L\alpha^2 d}{2}$$

To guarantee convergence, $\alpha \sim [0, \frac{\sqrt{2}\epsilon_g}{Ld}]$, then suppose $\alpha = \frac{\sqrt{2}\epsilon_g}{2Ld}$, we have $\mathbb{E}[f(\mathbf{x}_{k+1}) - f(\mathbf{x}_k)|\mathbf{x}_k] \leq -\frac{\epsilon_g^2}{4Ld}$ which completes the proof. $\square$

## B.2 QUERY COMPLEXITY OF S2P

**Theorem B.1.** (Theorem 3.1) (Query complexity) Suppose objective function $f(\cdot)$ satisfies Assumption 1. If we run algorithm 1 with step size strategy options 1 or 2, the algorithm returns in expectation an $\epsilon$-first-order stationary point in $\mathcal{O}(\frac{d}{\epsilon^2})$ function evaluations.

*Proof.* Using $L$-gradient Lipschitz, we have (descent lemma)

$$\mathbb{E}[f(\mathbf{x}_{k+1})|\mathbf{x}_k] \leq f(\mathbf{x}_k) + \mathbb{E}[\nabla f(\mathbf{x}_k)^T(\mathbf{x}_{k+1} - \mathbf{x}_k)|\mathbf{x}_k] + \frac{L}{2}\mathbb{E}[||\mathbf{x}_{k+1} - \mathbf{x}_k||^2]$$

$$= f(\mathbf{x}_k) - \alpha\mathbb{E}|\nabla f(\mathbf{x}_k)^T\mathbf{s}_k| + \frac{L\alpha^2}{2}\mathbb{E}||\mathbf{s}_k||_2^2 \quad \text{Take updating step}$$

$$= f(\mathbf{x}_k) - \alpha\mathbb{E}|\nabla f(\mathbf{x}_k)^T\mathbf{s}_k| + \frac{L\alpha^2 d}{2} \tag{3}$$

**Option 1. Stationary step size**

Lemma 2 shows that $\mathbb{E}_{\mathbf{s}_k\sim\mathcal{R}}|\nabla f(\mathbf{x}_k)^T\mathbf{s}_k| \geq \frac{1}{\sqrt{2}}||\nabla f(\mathbf{x}_k)||_2$, then inequality (3) can be reformulated as

$$\mathbb{E}[f(\mathbf{x}_{k+1})|\mathbf{x}_k] \leq f(\mathbf{x}_k) - \frac{\alpha}{\sqrt{2}}||\nabla f(\mathbf{x}_k)||_2 + \frac{L\alpha^2 d}{2}$$

Taking expectations in the above inequality w.r.t. $\mathbf{s}_k$ conditional on $\mathbf{x}_k$, and denoting $\theta_k = \mathbb{E}[f(\mathbf{x}_{k+1})]$ and $g_k = \mathbb{E}[||\nabla f(\mathbf{x}_k)||_2]$, we have

$$\theta_{k+1} \leq \theta_k - \frac{\alpha g_k}{\sqrt{2}} + \frac{L\alpha^2 d}{2}$$

$$g_k \leq \sqrt{2}(\frac{\theta_k - \theta_{k+1}}{\alpha} + \frac{L\alpha d}{2})$$

$$\sum_{k=0}^{K} g_k \leq \sqrt{2}(\frac{\theta_0 - \theta_{k+1}}{\alpha} + \frac{KL\alpha d}{4})$$

We can conclude that there exists an iteration $j \sim [0, K]$ such that

$$g_j \leq \sqrt{2}(\frac{\theta_0 - \theta_{k+1}}{\alpha K} + \frac{L\alpha d}{2})$$

$$g_j \leq \sqrt{2}(\frac{(f(\mathbf{x}_0) - f^\star)\sqrt{Kd}}{\alpha_0 K} + \frac{L\alpha_0 \sqrt{d}}{2\sqrt{K}}) \quad \text{By taking } \alpha = \frac{\alpha_0}{\sqrt{Kd}}$$

$$g_j \leq \frac{\sqrt{2d}}{\sqrt{K}}(\frac{(f(\mathbf{x}_0) - f^\star)}{\alpha_0} + \frac{L\alpha_0}{2})$$

Then let $\frac{\sqrt{2d}}{\sqrt{K}}(\frac{(f(\mathbf{x}_0)-f^\star)}{\alpha_0} + \frac{L\alpha_0}{2}) \leq \epsilon$, we have

$$K \geq \frac{2d}{\epsilon^2}(\frac{(f(\mathbf{x}_0) - f^\star)}{\alpha_0} + \frac{L\alpha_0}{2})^2,$$

, which completes the proof for option 1.

**Option 2. Dynamic step size**

Taking expectations in the above inequality (3) w.r.t. $\mathbf{s}_k$ conditional on $\mathbf{x}_k$, and denoting $\theta_k = \mathbb{E}[f(\mathbf{x}_{k+1})]$, we have

$$\theta_{k+1} \leq \theta_k - \alpha|\nabla f(\mathbf{x}_k)^T \mathbf{s}_k| + \frac{L\alpha^2 d}{2} \tag{4}$$

We know that the best $\alpha_k^{opt} = \frac{|\nabla f(\mathbf{x}_k)^T \mathbf{s}_k|}{Ld}$, and we can approximate the best step size with $\alpha_k = \frac{|f(\mathbf{x}+\rho\mathbf{s}_k)-f(\mathbf{x}-\rho\mathbf{s}_k)|}{2\rho Ld}$ (or $\alpha_k = \alpha_0 \frac{|f(\mathbf{x}+\rho\mathbf{s}_k)-f(\mathbf{x}-\rho\mathbf{s}_k)|}{2\rho}$ where $\alpha_0 = \frac{1}{Ld}$) where $\rho$ is a scalar.

Before continuing working on the inequality (4), we estimate the error between the best step size and the approximated step size, $|\delta_k| := |\alpha_k - \alpha_k^{opt}|$, firstly.

$$|\delta_k| = \frac{1}{2\rho Ld}\big||f(\mathbf{x}+\rho\mathbf{s}_k) - f(\mathbf{x}-\rho\mathbf{s}_k)| - 2\rho|\nabla f(\mathbf{x}_k)^T\mathbf{s}_k|\big|$$

$$\leq \frac{1}{2\rho Ld}|f(\mathbf{x}+\rho\mathbf{s}_k) - f(\mathbf{x}-\rho\mathbf{s}_k) - 2\rho\nabla f(\mathbf{x}_k)^T\mathbf{s}_k| \tag{5}$$

$$= \frac{1}{2\rho Ld}|(f(\mathbf{x}+\rho\mathbf{s}_k) - f(\mathbf{x}) - \rho\nabla f(\mathbf{x}_k)^T\mathbf{s}_k) - (f(\mathbf{x}-\rho\mathbf{s}_k) - f(\mathbf{x}) + \rho\nabla f(\mathbf{x}_k)^T\mathbf{s}_k)|$$

$$\leq \frac{1}{2\rho Ld}(\frac{L}{2}\rho^2\|\mathbf{s}_k\|^2 + \frac{L}{2}\rho^2\|\mathbf{s}_k\|^2) \tag{6}$$

$$\leq \frac{\rho}{2} \tag{7}$$

Note that inequality (5) applied reverse triangle inequality and inequality (6) applied the equivalent definitions of $L$-smooth function $|f(\mathbf{x}+\rho\mathbf{s}_k) - f(\mathbf{x}) - \rho\nabla f(\mathbf{x}_k)^T\mathbf{s}_k| \leq \frac{L}{2}\|\rho\mathbf{s}_k\|^2$.

Suppose we do take $\alpha_k = \frac{|f(\mathbf{x}+\rho\mathbf{s}_k)-f(\mathbf{x}-\rho\mathbf{s}_k)|}{2\rho Ld}$ and substitute $\alpha_k = \alpha_k^{opt} + \delta_k$, inequality (4) can be reformulated as

$$\theta_{k+1} \leq \theta_k - (\alpha_k^{opt} + \delta_k)|\nabla f(\mathbf{x}_k)^T\mathbf{s}_k| + \frac{L(\alpha_k^{opt} + \delta_k)^2 d}{2}$$

$$= \theta_k - \frac{|\nabla f(\mathbf{x}_k)^T\mathbf{s}_k|^2}{Ld} - \delta_k|\nabla f(\mathbf{x}_k)^T\mathbf{s}_k| + \frac{|\nabla f(\mathbf{x}_k)^T\mathbf{s}_k|^2}{2Ld} + \delta_k|\nabla f(\mathbf{x}_k)^T\mathbf{s}_k| + \frac{Ld\delta_k^2}{2}$$

$$= \theta_k - \frac{|\nabla f(\mathbf{x}_k)^T\mathbf{s}_k|^2}{2Ld} + \frac{Ld\delta_k^2}{2}$$

$$\leq \theta_k - \frac{|\nabla f(\mathbf{x}_k)^T\mathbf{s}_k|^2}{2Ld} + \frac{Ld\rho^2}{8} \quad \text{Apply inequality (7)}$$

$$\leq \theta_k - \frac{\|\nabla f(\mathbf{x}_k)\|^2}{4Ld} + \frac{Ld\rho^2}{8} \quad \text{Apply Lemma 2} \tag{8}$$

Note that it actually put requirement on $\rho$ to guarantee convergence, i.e., for $\rho_k$ in each iterations, we need $0 < \rho \leq \frac{\sqrt{2}\|\nabla f(\mathbf{x}_k)\|}{Ld}$.

Continually, inequality (8) further can be re-formulated as

$$||\nabla f(\mathbf{x}_k)||^2 \leq 4Ld(\theta_k - \theta_{k+1}) + \frac{\rho^2}{2}$$

$$\sum_{k=0}^{K} ||f(\mathbf{x}_k)||^2 \leq 4Ld(\theta_0 - \theta_{k+1}) + \frac{K\rho^2}{2}$$

We can conclude that there exists an iteration $j \sim [0, K]$ such that

$$||f(\mathbf{x}_j)||^2 \leq \frac{4Ld(\theta_0 - \theta_{k+1})}{K} + \frac{\rho^2}{2} \leq \frac{4Ld(f(\mathbf{x}_0) - f^\star)}{K} + \frac{\rho^2}{2}$$

which further concludes that we need

$$K \geq \frac{4Ld(f(\mathbf{x}_0) - f^\star)}{\epsilon^2 - \frac{\rho^2}{2}}, \tag{9}$$

iterations to reach $\epsilon$-first-order stationary point ($||f(\mathbf{x}_j)|| \leq \epsilon$).

Meanwhile, we require that $0 < \rho_k \leq \frac{\sqrt{2}||\nabla f(\mathbf{x}_k)||}{Ld}$ for $\rho_k$ in each iterations, and it can be set to a small value universally. E.g., $0 < \rho \leq \frac{\sqrt{2}\epsilon}{Ld}$, then we have $K \geq \frac{4Ld(f(\mathbf{x}_0) - f^\star)}{\epsilon^2(1 - \frac{1}{L^2 d^2})}$.

Then, we can safely conclude that the algorithm returns in expectation an $\epsilon$-first-order stationary point in $\mathcal{O}(\frac{d}{\epsilon^2})$ function evaluations, which completes the proof for option 2. □

## C  CONVERGENCE ANALYSIS UNDER THE RELAXED SMOOTHNESS ASSUMPTION

### C.1  PROGRESSIVE BOUND OF S2P

**Lemma C.1.** (Lemma 3.3) (Progressive bound) Suppose objective function $f(\cdot)$ satisfies Assumption 2 and $||\nabla f(\mathbf{x}_k)||_2 \geq \epsilon_g$. If we run algorithm 1 with step size $\alpha = \frac{\sqrt{2}\epsilon_g}{2(AL_0 + BL_1\epsilon_g)d}$, we have following progressive bound $\mathbb{E}[f(\mathbf{x}_{k+1}) - f(\mathbf{x}_k)|\mathbf{x}_k] \leq -\Omega(\frac{\epsilon_g^2}{(AL_0 + BL_1\epsilon_g)d})$, where $\mathbb{E}[\cdot|\mathbf{x}_k]$ denotes the conditional expectation w.r.t. $\mathbf{x}_k$, and constants $A = 1.01, B = 1.01$.

*Proof.* Give the decent lemma inequality (1), we have

$$\mathbb{E}[f(\mathbf{x}_{k+1})] \leq f(\mathbf{x}_k) - \alpha\mathbb{E}[\mathbf{g}_k^T\nabla f(\mathbf{x}_k)] + \frac{AL_0 + BL_1||\nabla f(\mathbf{x}_k)||}{2}\mathbb{E}[\alpha^2||\mathbf{g}_k||^2]$$

$$= f(\mathbf{x}_k) - \alpha\mathbb{E}[|\mathbf{s}_k^T\nabla f(\mathbf{x}_k)|] + \frac{AL_0 + BL_1||\nabla f(\mathbf{x}_k)||}{2}\mathbb{E}[\alpha^2||\mathbf{s}_k||^2] \quad \text{Take updating step}$$

$$\leq f(\mathbf{x}_k) - \frac{\alpha}{\sqrt{2}}||\nabla f(\mathbf{x}_k)|| + \alpha^2\frac{AL_0 + BL_1||\nabla f(\mathbf{x}_k)||}{2}d \quad \text{Lemma 2} \tag{10}$$

Suppose $||\nabla f(\mathbf{x}_k)|| \geq \epsilon_g$, and to guarantee convergence $\alpha \in [0, \frac{\sqrt{2}\epsilon_g}{(AL_0 + BL_1\epsilon_g)d}]$. Let $\alpha = \frac{\sqrt{2}\epsilon_g}{2(AL_0 + BL_1\epsilon_g)d}$, we have

$$\mathbb{E}[f(\mathbf{x}_{k+1})] \leq f(\mathbf{x}_k) - \frac{\epsilon_g^2}{4(AL_0 + BL_1\epsilon_g)d}.$$

which completes the proof.

Note that for the specific value of $A$ and $B$, we have $A = 1 + e^c - \frac{e^c - 1}{c}$, $B = \frac{e^c - 1}{c}$ and $||\mathbf{x}_{k+1} - \mathbf{x}_k|| = ||\alpha\mathbf{s}_k|| = \frac{\sqrt{2}\epsilon_g}{2(AL_0 + BL_1\epsilon_g)\sqrt{d}} \leq \frac{c}{L_1} \to c \geq \frac{\sqrt{2}L_1\epsilon_g}{2(AL_0 + BL_1\epsilon_g)\sqrt{d}} \to c \geq \frac{1}{\sqrt{2d}B} \to e^c \geq 1 + \frac{1}{\sqrt{2d}}$. It is easy to see that such $c$ exists, we can safely consider $A = 1.01, B = 1.01$ for simplicity (under large $d$) since $A$ and $B$ are expected to be small values. □

### C.2 QUERY COMPLEXITY OF S2P

**Theorem C.1.** (Theorem 3.2) (Query complexity) Suppose objective function $f(\cdot)$ satisfies Assumption 2. If we run algorithm 1 with step size strategy options 3 or 4, the algorithm returns in expectation an $\epsilon$-first-order stationary point in $\mathcal{O}(\frac{d}{\epsilon^2})$ function evaluations.

*Proof.* Give the decent lemma inequality (1), we have

$$
\begin{aligned}
\mathbb{E}[f(\mathbf{x}_{k+1})] &\leq f(\mathbf{x}_k) - \alpha \mathbb{E}[\mathbf{g}_k^T \nabla f(\mathbf{x}_k)] + \frac{AL_0 + BL_1||\nabla f(\mathbf{x}_k)||}{2} \mathbb{E}[\alpha^2 ||\mathbf{g}_k||^2] \\
&= f(\mathbf{x}_k) - \alpha \mathbb{E}[|\mathbf{s}_k^T \nabla f(\mathbf{x}_k)|] + \alpha^2 \frac{AL_0 + BL_1||\nabla f(\mathbf{x}_k)||}{2} \mathbb{E}[||\mathbf{s}_k||^2] \quad \text{Take updating step}
\end{aligned}
$$
(11)

**Option 1. Stationary step size**

Lemma 2 shows that $\mathbb{E}_{\mathbf{s}_k \sim \mathcal{R}} |\nabla f(\mathbf{x}_k)^T \mathbf{s}_k| \geq \frac{1}{\sqrt{2}} ||\nabla f(\mathbf{x}_k)||_2$, then inequality (11) can be reformulated as

$$
\mathbb{E}[f(\mathbf{x}_{k+1})] \leq f(\mathbf{x}_k) - \frac{\alpha}{\sqrt{2}} ||\nabla f(\mathbf{x}_k)|| + \alpha^2 \frac{AL_0 + BL_1||\nabla f(\mathbf{x}_k)||}{2} d
$$

Taking expectations in the above inequality w.r.t. $\mathbf{s}_k$ conditional on $\mathbf{x}_k$, and denoting $\theta_k = \mathbb{E}[f(\mathbf{x}_{k+1})]$ and $g_k = \mathbb{E}[||\nabla f(\mathbf{x}_k)||]$, we have

$$
\theta_{k+1} \leq \theta_k - \frac{\alpha}{\sqrt{2}} g_k + \alpha^2 \frac{AL_0 + BL_1 g_k}{2} d
$$

$$
g_k \left( \frac{\sqrt{2}\alpha - B\alpha^2 L_1 d}{2} \right) \leq \theta_k - \theta_{k+1} + \frac{A\alpha^2 L_0 d}{2}
$$

$$
g_k \leq \frac{2(\theta_k - \theta_{k+1})}{\sqrt{2}\alpha - B\alpha^2 L_1 d} + \frac{A\alpha^2 L_0 d}{\sqrt{2}\alpha - B\alpha^2 L_1 d}
$$

$$
\sum_{k=0}^{K} g_k \leq \frac{2(\theta_0 - \theta_{k+1})}{\sqrt{2}\alpha - B\alpha^2 L_1 d} + \frac{K A\alpha^2 L_0 d}{\sqrt{2}\alpha - B\alpha^2 L_1 d}
$$

We can conclude that there exists an iteration $j \sim [0, K]$ such that

$$
\begin{aligned}
g_j &\leq \frac{2(\theta_0 - \theta_{K+1})}{(\sqrt{2}\alpha - B\alpha^2 L_1 d)K} + \frac{A\alpha^2 L_0 d}{\sqrt{2}\alpha - B\alpha^2 L_1 d} \\
&\leq \frac{2(f(\mathbf{x}_0) - f^\star)}{(\sqrt{2}\alpha - B\alpha^2 L_1 d)K} + \frac{A\alpha^2 L_0 d}{\sqrt{2}\alpha - B\alpha^2 L_1 d}
\end{aligned}
$$
(12)

Suppose $\alpha = \frac{\sqrt{2}}{BL_1\sqrt{dK}}$, inequality (12) can be reformulated as

$$
g_j \leq \frac{B(f(\mathbf{x}_0) - f^\star)L_1\sqrt{d}}{\sqrt{K} - \sqrt{d}} + \frac{AL_0\sqrt{d}}{BL_1(\sqrt{K} - \sqrt{d})}.
$$

Under this setting, we can see that the $g_j$ can be continually decreased with at least $K > d$, which further shows that it need

$$
K \geq (\sqrt{d} + \frac{AL_0\sqrt{d} + BL_1(f(\mathbf{x}_0) - f^\star)\sqrt{d}}{\epsilon})^2
$$

iterations to reach $\epsilon$-first-order stationary point. Then, we can safely conclude that the algorithm returns in expectation an $\epsilon$-first-order stationary point in $\mathcal{O}(\frac{d}{\epsilon^2})$ function evaluations, which completes the proof for option 1.

Note that for the specific value of $A$ and $B$, we have $A = 1 + e^c - \frac{e^c - 1}{c}$, $B = \frac{e^c - 1}{c}$ and $||\mathbf{x}_{k+1} - \mathbf{x}_k|| = ||\alpha \mathbf{s}_k|| = \frac{\sqrt{2}}{BL_1\sqrt{K}} \leq \frac{c}{L_1} \rightarrow c \geq \frac{\sqrt{2}}{B\sqrt{K}} \rightarrow e^c \geq 1 + \sqrt{\frac{2}{K}}$. It is easy to see that such $c$ exists, we can

safely consider $A = 1.01, B = 1.01$ for simplicity (under large $d$) since $A$ and $B$ are expected to be small values.

### Option 2. Dynamic step size

Taking expectations in the above inequality (11) w.r.t. $\mathbf{s}_k$ conditional on $\mathbf{x}_k$, and denoting $\theta_k = \mathbb{E}[f(\mathbf{x}_{k+1})]$, we have

$$\theta_{k+1} \leq \theta_k - \alpha|\mathbf{s}_k^T \nabla f(\mathbf{x}_k)| + \alpha^2 \frac{AL_0 + BL_1||\nabla f(\mathbf{x}_k)||}{2}d$$

$$\leq \theta_k - \alpha|\mathbf{s}_k^T \nabla f(\mathbf{x}_k)| + \alpha^2 \frac{AL_0 + \sqrt{2}BL_1|\mathbf{s}_k^T \nabla f(\mathbf{x}_k)|}{2}d. \tag{13}$$

It is easy to know that $\alpha_k^{opt} = \frac{|\mathbf{s}^T \nabla f(\mathbf{x}_k)|}{(AL_0 + \sqrt{2}BL_1|\mathbf{s}^T \nabla f(\mathbf{x}_k)|)d}$. Let $|\gamma_k| = \frac{|f(\mathbf{x}_k + \rho \mathbf{s}_k) - f(\mathbf{x}_k - \rho \mathbf{s}_k)|}{2\rho}$, and we approximate the best step size with $\alpha_k = \frac{|\gamma_k|}{(AL_0 + \sqrt{2}BL_1|\gamma_k|)d}$ and denote the approximation error as $|\delta_k| := |\alpha_k - \alpha_k^{opt}|$.

Before we continue working on the inequality (13), we derive the upper bound of $|\delta_k|$ for our following analysis. Firstly, we denote $|\epsilon_\rho| := \left||\mathbf{s}^T \nabla f(\mathbf{x}_k)| - |\gamma_k|\right| = \left||\mathbf{s}^T \nabla f(\mathbf{x}_k)| - \frac{|f(\mathbf{x}_k + \rho \mathbf{s}_k) - f(\mathbf{x}_k - \rho \mathbf{s}_k)|}{2\rho}\right| = \mathcal{O}(\rho^2 d^{3/2})$ (Taylor expansion). So that, we can define $|\epsilon_\rho| \leq \xi \rho^2 d^{3/2}$ where $\xi$ is a constant associated with third-order property of $f$. Note $d^{3/2}$ is the compensation of normalizing $\mathbf{s}$.

Specifically, we try to prove $|\delta_k| \leq |\epsilon_\rho|$. We define a new function $g(x) = \frac{x}{AL_0 + \sqrt{2}BL_1 x}$, then to prove $|\delta_k| \leq |\epsilon_\rho|$ is equivalent to prove $|g(|\mathbf{s}^T \nabla f(\mathbf{x}_k)|) - g(|\gamma_k|)| \leq d\left||\mathbf{s}^T \nabla f(\mathbf{x}_k)| - |\gamma_k|\right|$, further it is equivalent to prove $g'(x) = \frac{AL_0}{(AL_0 + \sqrt{2}BL_1 x)} \leq d$ when $x \geq 0$, which is obviously true. Overall, we have approximation error $|\delta_k| \leq \xi \rho^2 d^{3/2}$.

Then, we continue our analysis. Suppose we do take step size $\alpha_k = \frac{|\gamma_k|}{(AL_0 + \sqrt{2}BL_1|\gamma_k|)d}$ and substitute $\alpha_k = \alpha_k^{opt} + \delta_k$, then inequality (13) can be re-formulate as

$$\theta_{k+1} \leq \theta_k - (\alpha_k^{opt} + \delta_k)|\mathbf{s}_k^T \nabla f(\mathbf{x}_k)| + (\alpha_k^{opt} + \delta_k)^2 \frac{AL_0 + \sqrt{2}BL_1|\mathbf{s}_k^T \nabla f(\mathbf{x}_k)|}{2}d$$

$$= \theta_k - \frac{||\mathbf{s}^T \nabla f(\mathbf{x}_k)||^2}{(AL_0 + \sqrt{2}BL_1|\mathbf{s}^T \nabla f(\mathbf{x}_k)|)d} - |\mathbf{s}^T \nabla f(\mathbf{x}_k)|\delta_k + \frac{||\mathbf{s}^T \nabla f(\mathbf{x}_k)||^2}{2(AL_0 + \sqrt{2}BL_1|\mathbf{s}^T \nabla f(\mathbf{x}_k)|)d}$$

$$+ \frac{AL_0 + \sqrt{2}BL_1|\mathbf{s}_k^T \nabla f(\mathbf{x}_k)|}{2}d\delta_k^2 + |\mathbf{s}^T \nabla f(\mathbf{x}_k)|\delta_k$$

$$\leq \theta_k - \frac{||\mathbf{s}^T \nabla f(\mathbf{x}_k)||^2}{2(AL_0 + \sqrt{2}BL_1|\mathbf{s}^T \nabla f(\mathbf{x}_k)|)d} + \frac{(AL_0 + \sqrt{2}BL_1|\mathbf{s}_k^T \nabla f(\mathbf{x}_k)|)d}{2}\delta_k^2$$

$$\leq \theta_k - \frac{||\nabla f(\mathbf{x}_k)||^2}{4(AL_0 + \sqrt{2}BL_1||\nabla f(\mathbf{x}_k)||)d} + \frac{(AL_0 + \sqrt{2}BL_1||\nabla f(\mathbf{x}_k)||)d}{2}\delta_k^2 \quad \text{Apply Lemma 2}$$
$$\tag{14}$$

### Condition 1

Suppose $1 - \sqrt{2}BL_1 \geq 0$ and $||\nabla f(\mathbf{x}_k)|| \geq AL_0 + \sqrt{2}BL_1||\nabla f(\mathbf{x}_k)||$, inequality (14) can be reformulated as

$$\theta_{k+1} \leq \theta_k - \frac{||\nabla f(\mathbf{x}_k)||}{4d} + \frac{||\nabla f(\mathbf{x}_k)||d}{2}\delta_k^2$$

Meanwhile, suppose $|\delta_k| \leq \xi \rho^2 d^{3/2} \leq \frac{1}{2d}$, we have

$$||\nabla f(\mathbf{x}_k)|| \leq 8d(\theta_k - \theta_{k+1})$$

$$\sum_{k=0}^{K} ||\nabla f(\mathbf{x}_k)|| \leq 8d(\theta_0 - \theta_{k+1})$$

We can conclude that there exists an iteration $j \sim [0, K]$ such that

$$||\nabla f(\mathbf{x}_j)|| \leq \frac{8d(\theta_0 - \theta_{k+1})}{K}$$

$$||\nabla f(\mathbf{x}_j)|| \leq \frac{8d(f(\mathbf{x}_0) - f^\star)}{K}$$

which concludes that we need

$$K \geq \frac{8d(f(\mathbf{x}_0) - f^\star)}{\epsilon} \tag{15}$$

iterations to reach $\epsilon$-first-order stationary point.

**Condition 2**

Suppose $1 - \sqrt{2}BL_1 \geq 0$ and $||\nabla f(\mathbf{x}_k)|| \leq AL_0 + \sqrt{2}BL_1||\nabla f(\mathbf{x}_k)||$, we have $||\nabla f(\mathbf{x}_k)|| \leq \frac{AL_0}{1-\sqrt{2}BL_1}$. Meanwhile, suppose $|\delta_k| \leq \xi\rho^2 d^{3/2} \leq \frac{||\nabla f(\mathbf{x}_k)||}{2(AL_0+\sqrt{2}BL_1||\nabla f(\mathbf{x}_k)||)d}$, then inequality (14) can be reformulated as

$$\theta_{k+1} \leq \theta_k - \frac{||\nabla f(\mathbf{x}_k)||^2}{8(AL_0 + \sqrt{2}BL_1\frac{AL_0}{1-\sqrt{2}BL_1})d}$$

$$||\nabla f(\mathbf{x}_k)||^2 \leq (\theta_k - \theta_{k+1})\frac{8AL_0 d}{1 - \sqrt{2}BL_1}$$

$$\sum_{k=0}^{K} ||\nabla f(\mathbf{x}_k)||^2 \leq (\theta_0 - \theta_{k+1})\frac{8AL_0 d}{1 - \sqrt{2}BL_1}$$

We can conclude that there exists an iteration $j \sim [0, K]$ such that

$$||\nabla f(\mathbf{x}_j)||^2 \leq \frac{8AL_0 d(\theta_0 - \theta_{k+1})}{(1 - \sqrt{2}BL_1)K}$$

$$||\nabla f(\mathbf{x}_j)|| \leq \sqrt{\frac{8AL_0 d(f(\mathbf{x}_0) - f^\star)}{(1 - \sqrt{2}BL_1)K}},$$

which concludes that we need

$$K \geq \frac{8AL_0 d(f(\mathbf{x}_0) - f^\star)}{(1 - \sqrt{2}BL_1)\epsilon^2}$$

iterations to reach $\epsilon$-first-order stationary point.

**Condition 3**

Suppose $1 - \sqrt{2}BL_1 \leq 0$ and $||\nabla f(\mathbf{x}_k)||^2 \leq (\frac{AL_0}{1-\sqrt{2}BL_1})^2$. Meanwhile, suppose $|\delta_k| \leq \xi\rho^2 d^{3/2} \leq \frac{||\nabla f(\mathbf{x}_k)||}{2(AL_0+\sqrt{2}BL_1||\nabla f(\mathbf{x}_k)||)d}$, then inequality (14) can be reformulated as

$$\theta_{k+1} \leq \theta_k - \frac{||\nabla f(\mathbf{x}_k)||^2}{8(AL_0 + \sqrt{2}BL_1|\frac{AL_0}{1-\sqrt{2}BL_1}|)d}$$

$$||\nabla f(\mathbf{x}_k)||^2 \leq (\theta_k - \theta_{k+1})\frac{8AL_0 d(2\sqrt{2}BL_1 - 1)}{\sqrt{2}BL_1 - 1}$$

$$\sum_{k=0}^{K} ||\nabla f(\mathbf{x}_k)||^2 \leq (\theta_0 - \theta_{k+1})\frac{8AL_0 d(2\sqrt{2}BL_1 - 1)}{\sqrt{2}BL_1 - 1}$$

We can conclude that there exists an iteration $j \sim [0, K]$ such that

$$||\nabla f(\mathbf{x}_j)||^2 \leq \frac{8AL_0 d(\theta_0 - \theta_{k+1})(2\sqrt{2}BL_1 - 1)}{(\sqrt{2}BL_1 - 1)K}$$

$$||\nabla f(\mathbf{x}_j)|| \leq \sqrt{\frac{8AL_0 d(f(\mathbf{x}_0) - f^\star)(2\sqrt{2}BL_1 - 1)}{(\sqrt{2}BL_1 - 1)K}},$$

which concludes that we need

$$K \geq \frac{8AL_0 d(f(\mathbf{x}_0) - f^\star)(2\sqrt{2}BL_1 - 1)}{(\sqrt{2}BL_1 - 1)\epsilon^2} \tag{16}$$

iterations to reach $\epsilon$-first-order stationary point.

**Condition 4**

Suppose $1 - \sqrt{2}BL_1 \leq 0$ and $||\nabla f(\mathbf{x}_k)||^2 \geq (\frac{AL_0}{1-\sqrt{2}BL_1})^2$. Meanwhile, suppose $\delta_k \leq \xi\rho^2 d^{3/2} \leq \frac{||\nabla f(\mathbf{x}_k)||}{2(AL_0 + \sqrt{2}BL_1||\nabla f(\mathbf{x}_k)||)d}$, then inequality (14) can be reformulated as

$$\theta_{k+1} \leq \theta_k - \frac{(\frac{AL_0}{1-\sqrt{2}BL_1})^2}{8(AL_0 + \sqrt{2}BL_1||\nabla f(\mathbf{x}_k)||)d} \tag{17}$$

Since $\frac{(\frac{AL_0}{1-\sqrt{2}BL_1})^2}{8(AL_0 + \sqrt{2}BL_1||\nabla f(\mathbf{x}_k)||)d}$ is a monotone decreasing function w.r.t. $||\nabla f(\mathbf{x}_k)||$, then we can conclude that the loss function cannot be indicator of reaching $\epsilon$-first-order stationary points. However, with an appropriate selection of parameters, the loss function can be minimized. I.e.,

$$\theta_{k+1} \leq \theta_k - \frac{(\frac{AL_0}{\sqrt{2}BL_1-1})^2}{8(AL_0 + \sqrt{2}BL_1\frac{AL_0}{\sqrt{2}BL_1-1})d}$$

$$\theta_{k+1} \leq \theta_k - \frac{AL_0}{8(2\sqrt{2}BL_1 - 1)(\sqrt{2}BL_1 - 1)d}$$

$$\theta_{k+1} \leq \theta_0 - (K+1)\frac{AL_0}{8(2\sqrt{2}BL_1 - 1)(\sqrt{2}BL_1 - 1)d}$$

$$f(\mathbf{x}_k) - f^\star \leq f(\mathbf{x}_0) - f^\star - K\frac{AL_0}{8(2\sqrt{2}BL_1 - 1)(\sqrt{2}BL_1 - 1)d},$$

which concludes that we need

$$K \geq \frac{8(2\sqrt{2}BL_1 - 1)(\sqrt{2}BL_1 - 1)(f(\mathbf{x}_0) - f^\star - \epsilon)d}{AL_0}$$

iterations to reach local $\epsilon$-optimal point.

We summarize the results over all conditions in Table 2.

| Conditions[b] | requirement over $\rho$[a] | Query complexity |
|---|---|---|
| $L_1 \leq \frac{1}{\sqrt{2}B}, ||\nabla f(\mathbf{x})|| \geq \frac{AL_0}{1-\sqrt{2}BL_1}$ | $\rho \leq \frac{1}{d\sqrt{2\xi\sqrt{d}}}$ | $\frac{8d(f(\mathbf{x}_0)-f^\star)}{\epsilon}$ |
| $L_1 \leq \frac{1}{\sqrt{2}B}, ||\nabla f(\mathbf{x})|| \leq \frac{AL_0}{1-\sqrt{2}BL_1}$ | $\rho \leq \frac{1}{d}\sqrt{\frac{\epsilon}{2\xi(AL_0+\sqrt{2}BL_1\epsilon)\sqrt{d}}}$ | $\frac{8AL_0 d(f(\mathbf{x}_0)-f^\star)}{(1-\sqrt{2}BL_1)\epsilon^2}$ |
| $L_1 \geq \frac{1}{\sqrt{2}B}, ||\nabla f(\mathbf{x})|| \leq \frac{AL_0}{\sqrt{2}BL_1-1}$ | $\rho \leq \frac{1}{d}\sqrt{\frac{\epsilon}{2\xi(AL_0+\sqrt{2}BL_1\epsilon)\sqrt{d}}}$ | $\frac{8AL_0 d(f(\mathbf{x}_0)-f^\star)(2\sqrt{2}BL_1-1)}{(\sqrt{2}BL_1-1)\epsilon^2}$ |
| $L_1 \geq \frac{1}{\sqrt{2}B}, ||\nabla f(\mathbf{x})|| \geq \frac{AL_0}{\sqrt{2}BL_1-1}$ | $\rho \leq \frac{1}{d}\sqrt{\frac{\epsilon}{2\xi(AL_0+\sqrt{2}BL_1\epsilon)\sqrt{d}}}$ | $\frac{8(2\sqrt{2}BL_1-1)(\sqrt{2}BL_1-1)(f(\mathbf{x}_0)-f^\star-\epsilon)d}{AL_0}$ |

[a] $\xi$ is a constant associated with third-order property of $f$, detailed in appendix inequality (13).
[b] For forth condition, reaching local $\epsilon$-optimal point instead of $\epsilon$-first-order stationary point, detailed in appendix inequality (17).

Table 2: With dynamic step size strategy, the convergence property of $f$ under relaxed smoothness.

Note that for the specific value of $A$ and $B$, we have $A = 1+e^c - \frac{e^c-1}{c}$, $B = \frac{e^c-1}{c}$ and $||\mathbf{x}_{k+1}-\mathbf{x}_k|| = ||\alpha\mathbf{s}_k|| = \frac{\gamma_k}{(AL_0+\sqrt{2}BL_1\gamma_k)\sqrt{d}} \leq \frac{c}{L_1} \to c \geq \frac{1}{B\sqrt{2d}} \to e^c \geq 1 + \frac{1}{\sqrt{2d}}$. It is easy to see that such $c$ exists, we can safely consider $A = 1.01, B = 1.01$ for simplicity (under large $d$) since $A$ and $B$ are expected to be small values. $\qquad\square$

## C.3 Bound of gradient norm of S2P

**Theorem C.2.** (Theorem 3.3) Suppose objective function $f(\cdot)$ satisfies Assumption 2. Then the gradient norm $||\nabla f(\mathbf{x}_k)||$ can be bounded in expectation as

$$|\gamma| - \rho d(AL_0 + BL_1||\nabla f(\mathbf{x})||) \leq ||\nabla f(\mathbf{x})|| \leq \sqrt{2}|\gamma| + \sqrt{2}\rho d(AL_0 + BL_1||\nabla f(\mathbf{x})||)$$

where $|\gamma| = \frac{|f(\mathbf{x}+\rho\mathbf{s}) - f(\mathbf{x}+\rho\mathbf{s})|}{2\rho}$. Constants $A = 1.01, B = 1.01$ when $\rho \leq \frac{0.001}{2L_1\sqrt{d}}$

*Proof.*

$$||\nabla f(\mathbf{x})|| \leq \mathbb{E}[\sqrt{2}|\mathbf{s}^T\nabla f(x)|] = \mathbb{E}[\frac{1}{\sqrt{2}\rho}|2\rho\mathbf{s}^T\nabla f(x)|] \tag{18}$$

$$= \mathbb{E}[\frac{1}{\sqrt{2}\rho}|(f(\mathbf{x}+\rho\mathbf{s}) - f(\mathbf{x}-\rho\mathbf{s})) - (f(\mathbf{x}+\rho\mathbf{s}) - f(\mathbf{x}-\rho\mathbf{s}) - 2\rho\mathbf{s}^T\nabla f(x))|]$$

$$\leq \mathbb{E}[\sqrt{2}\frac{|f(\mathbf{x}+\rho\mathbf{s}) - f(\mathbf{x}-\rho\mathbf{s})|}{2\rho} + \frac{1}{\sqrt{2}\rho}|f(\mathbf{x}+\rho\mathbf{s}) - f(\mathbf{x}-\rho\mathbf{s}) - 2\rho\mathbf{s}^T\nabla f(x))|]$$

$$= \sqrt{2}|\gamma| + \frac{1}{\sqrt{2}\rho}\mathbb{E}[|f(\mathbf{x}+\rho\mathbf{s}) - f(\mathbf{x}-\rho\mathbf{s}) - 2\rho\mathbf{s}^T\nabla f(x))|]$$

$$\leq \sqrt{2}|\gamma| + \frac{1}{\sqrt{2}\rho}\frac{AL_0 + BL_1||\nabla f(\mathbf{x})||}{2}\mathbb{E}[||2\rho\mathbf{s}||^2] \tag{19}$$

$$= \sqrt{2}|\gamma| + \sqrt{2}\rho d(AL_0 + BL_1||\nabla f(\mathbf{x})||).$$

Note inequality (18) applies Lemma A.2, inequality (19) applies Lemma A.1. And the same with the following proof.

$$||\nabla f(\mathbf{x})|| \geq \mathbb{E}[|\mathbf{s}^T\nabla f(x)|] = \mathbb{E}[\frac{1}{2\rho}|2\rho\mathbf{s}^T\nabla f(x)|]$$

$$= \mathbb{E}[\frac{1}{2\rho}|(f(\mathbf{x}+\rho\mathbf{s}) - f(\mathbf{x}-\rho\mathbf{s})) - (f(\mathbf{x}+\rho\mathbf{s}) - f(\mathbf{x}-\rho\mathbf{s}) - 2\rho\mathbf{s}^T\nabla f(x))|]$$

$$\geq \mathbb{E}[\frac{|f(\mathbf{x}+\rho\mathbf{s}) - f(\mathbf{x}-\rho\mathbf{s})|}{2\rho} - \frac{1}{2\rho}|f(\mathbf{x}+\rho\mathbf{s}) - f(\mathbf{x}-\rho\mathbf{s}) - 2\rho\mathbf{s}^T\nabla f(x))|]$$

$$= |\gamma| - \frac{1}{2\rho}\mathbb{E}[|f(\mathbf{x}+\rho\mathbf{s}) - f(\mathbf{x}-\rho\mathbf{s}) - 2\rho\mathbf{s}^T\nabla f(x))|]$$

$$\geq |\gamma| - \frac{1}{2\rho}\frac{AL_0 + BL_1||\nabla f(\mathbf{x})||}{2}\mathbb{E}[||2\rho\mathbf{s}||^2]$$

$$\geq |\gamma| - \rho d(AL_0 + BL_1||\nabla f(\mathbf{x})||).$$

Note that for the specific value of $A$ and $B$, we have $A = 1+e^c - \frac{e^c-1}{c}, B = \frac{e^c-1}{c}$ and $||\mathbf{x}_{k+1}-\mathbf{x}_k|| = ||(\mathbf{x}+\rho\mathbf{s}) - (\mathbf{x}-\rho\mathbf{s})|| = ||2\rho\mathbf{s}|| = 2\rho\sqrt{d} \leq \frac{c}{L_1} \rightarrow c \geq 2\rho L_1\sqrt{d}$. It is easy to see that such $c$ exists, we can safely consider $\rho \leq \frac{1}{2L_1d}$, then we have $c \geq \frac{1}{\sqrt{d}}$. It is easy to see such $c$ exists, we set $A = 1.01, B = 1.01$ for simplicity. $\square$

# D EXPERIMENTS

## D.1 SETUP

For experiment over common deep models and datasets, we do grid search for initial learning rate $\alpha_0$ over list {2e-4, 1e-4, 8e-5, 5e-5, 2e-5, 1e-5} and for smoothing parameter $\rho_0$ over list {1e-3, 5e-4, 1e-4, 5e-5, 1e-5} with all methods. We average the results across 5 random seeds.

Note the selected hyper-parameters directly apply to sign variants. The tunable hyper-parameters are summarized in Table 3.

| Hyper-parameter | Arc.&Dataset | Method | | |
|---|---|---|---|---|
| | | GA | AS2P | STP |
| $\alpha_0$ | ResNet18&CIFAR10 | 2.0e-5 | - | 2.0e-4 |
| | ResNet50&CIFAR10 | 1.0e-5 | - | 2.0e-4 |
| | ResNet101&CIFAR100 | 2.0e-5 | - | 1.0e-4 |
| | ResNet152&CIFAR100 | 2.0e-5 | - | 1.0e-4 |
| LR scheduler | All | Cosine decay | | |
| $\rho_0$ | ResNet18&CIFAR10 | 1e-3 | 1e-3 | - |
| | ResNet50&CIFAR10 | 1e-3 | 5e-4 | - |
| | ResNet101&CIFAR100 | 5e-4 | 5e-4 | - |
| | ResNet152&CIFAR100 | 5e-4 | 5e-4 | - |
| $\rho_{\text{end}}$ | All | - | $\rho_0/10$ | - |
| $\eta_a$ | All | - | 5 | - |
| $\eta_b$ | ResNet18&CIFAR10 | - | 5 | - |
| | ResNet50&CIFAR10 | - | 5 | - |
| | ResNet101&CIFAR100 | - | 3 | - |
| | ResNet152&CIFAR100 | - | 5 | - |
| Std Dev($\gamma_{\text{recent}}$) | All | - | 10% | - |

Table 3: Summary of hyper-parameters used in experiments over common deep models and datasets. It shows that AS2P has extra hyper-parameters $\rho_{\text{end}}$, $\eta_a$, $\eta_b$, and Std Dev($\gamma_{\text{recent}}$). Basically, those hyper-parameters are unnecessary to tune within above deep models and datasets.

For the experiment over LLM, the six text tasks follow the original settings exactly (Malladi et al., 2023), which randomly samples 1,000 examples and 500 examples for training and validation respectively for each task. We get the results with a fixed random seed. Specifically, for the learning rate and smoothing parameter, we apply the best values mentioned in Malladi et al. (2023) for GA. Then, AS2P directly applies the value of smoothing parameter $\rho_0$ from GA and only needs to tune one hyper-parameter $\eta_b$. For STP method, we search the best $\alpha_0$ from list {5e-5, 2e-5, 1e-5, 5e-6 ,1e-6, 1e-7}. The details of hyper-parameters are summarized in Table 4, which shows that only $\eta_b$ is necessary to update among all four extra hyper-parameters $\rho_{\text{end}}$, $\eta_a$, $\eta_b$, and Std Dev($\gamma_{\text{recent}}$) of AS2P compared with experiments about common deep models&datasets.

| Hyper-parameter | Task | Method | | | |
|---|---|---|---|---|---|
| | | GA | GA constant | AS2P | STP |
| $\alpha_0$ | SST-2 RTE Copa ReCoRD SQuAD DROP | 1e-7 | 1e-7 | - | 2e-5 |
| LR scheduler | All | Cosine decay | Constant value | Cosine decay | Cosine decay |
| $\rho_0$ | All | 1e-3 | 1e-3 | 1e-3 | - |
| $\rho_{\text{end}}$ | All | - | - | $\rho_0/10$ | - |
| $\eta_a$ | All | - | - | 5 | - |
| $\eta_b$ | All | - | - | 50 | - |
| Std Dev($\gamma_{\text{recent}}$) | All | - | - | 10% | - |

Table 4: Summary of hyper-parameters used in experiments over LLM. Basically, AS2P needs to tune $\eta_b$, and the selected values are robust across varying tasks.

## D.2 ADDITIONAL EXPERIMENTS

Table version of Figure 1(a) and Figure 2(a). The base of training cost ratio, e.g., {1, 0.8, 0.6, 0.4, 0.2}, normalizes the number of function queries when base method GA reaches {500, 400, 300, 200, 100} epochs with some specific loss values. Then, the training cost ratio aligns with the ratio between the number of function queries of the base method and other methods reaching the same loss values.

| Task | Method | Training cost ratio | | | | |
|---|---|---|---|---|---|---|
| ResNet18&CIFAR10 | GA | 1 | 0.80 | 0.60 | 0.40 | 0.20 |
| | STP | 1 | 1 | 0.80 | 0.43 | 0.17 |
| | AS2P | 0.56 | 0.52 | 0.39 | 0.24 | 0.10 |
| ResNet50&CIFAR10 | GA | 1 | 0.80 | 0.60 | 0.40 | 0.20 |
| | STP | 0.98 | 0.87 | 0.54 | 0.14 | 0.05 |
| | AS2P | 0.40 | 0.36 | 0.27 | 0.14 | 0.05 |
| ResNet101&CIFAR100 | GA | 1 | 0.80 | 0.60 | 0.40 | 0.20 |
| | STP | 0.93 | 0.85 | 0.61 | 0.37 | 0.14 |
| | AS2P | 0.39 | 0.36 | 0.28 | 0.17 | 0.07 |
| ResNet152&CIFAR100 | GA | 1 | 0.80 | 0.60 | 0.40 | 0.20 |
| | STP | 0.63 | 0.59 | 0.45 | 0.25 | 0.10 |
| | AS2P | 0.32 | 0.31 | 0.24 | 0.14 | 0.06 |

Table 5: Training cost ratio of reaching specific loss values under common deep models&datasets.

| Task | Method | Training cost ratio | | | |
|---|---|---|---|---|---|
| SST-2 | GA cosine decay LR | 1 | 0.75 | 0.50 | 0.25 |
| | GA constant LR | 0.47 | 0.50 | 0.38 | 0.23 |
| | STP cosine decay LR | 0.38 | 0.38 | 0.30 | 0.19 |
| | AS2P cosine decay LR | 0.17 | 0.17 | 0.15 | 0.10 |
| RTE | GA cosine decay LR | 1 | 0.75 | 0.50 | 0.25 |
| | GA constant LR | 0.35 | 0.35 | 0.30 | 0.12 |
| | STP cosine decay LR | 0.03 | 0.03 | 0.03 | 0.03 |
| | AS2P cosine decay LR | 0.03 | 0.03 | 0.03 | 0.03 |
| Copa | GA cosine decay LR | 1 | 0.75 | 0.50 | 0.25 |
| | GA constant LR | 0.47 | 0.47 | 0.40 | 0.23 |
| | STP cosine decay LR | 0.26 | 0.26 | 0.23 | 0.11 |
| | AS2P cosine decay LR | 0.10 | 0.10 | 0.07 | 0.05 |
| ReCoRD | GA cosine decay LR | 1 | 0.75 | 0.50 | 0.25 |
| | GA constant LR | 0.42 | 0.42 | 0.42 | 0.23 |
| | STP cosine decay LR | 0.23 | 0.23 | 0.19 | 0.07 |
| | AS2P cosine decay LR | 0.05 | 0.05 | 0.05 | 0.03 |
| SQuAD | GA cosine decay LR | 1 | 0.75 | 0.50 | 0.25 |
| | GA constant LR | 0.65 | 0.65 | 0.53 | 0.30 |
| | STP cosine decay LR | 0.23 | 0.23 | 0.23 | 0.15 |
| | AS2P cosine decay LR | 0.15 | 0.15 | 0.12 | 0.07 |
| DROP | GA cosine decay LR | 1 | 0.75 | 0.50 | 0.25 |
| | GA constant LR | 0.42 | 0.47 | 0.40 | 0.23 |
| | STP cosine decay LR | 0.34 | 0.34 | 0.30 | 0.19 |
| | AS2P cosine decay LR | 0.15 | 0.17 | 0.15 | 0.10 |

Table 6: Training cost ratio of reaching specific loss values when fully fine-tuning OPT-13B model under various tasks.

Figure 4: Performance comparison between applying different noise distributions such as Normal distribution, Rademacher distribution, and Uniform distribution.

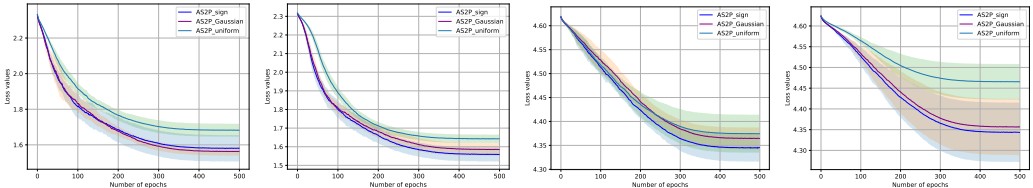

(a) Under pre-trained ResNet18&CIFAR10   (b) Under pre-trained ResNet50&CIFAR10   (c) Under pre-trained ResNet101&CIFAR100   (d) Under pre-trained ResNet152&CIFAR100

Figure 5: Convergence rate of pre-trained ResNet18&CIFAR100 and pre-trained ResNet50&CIFAR100.

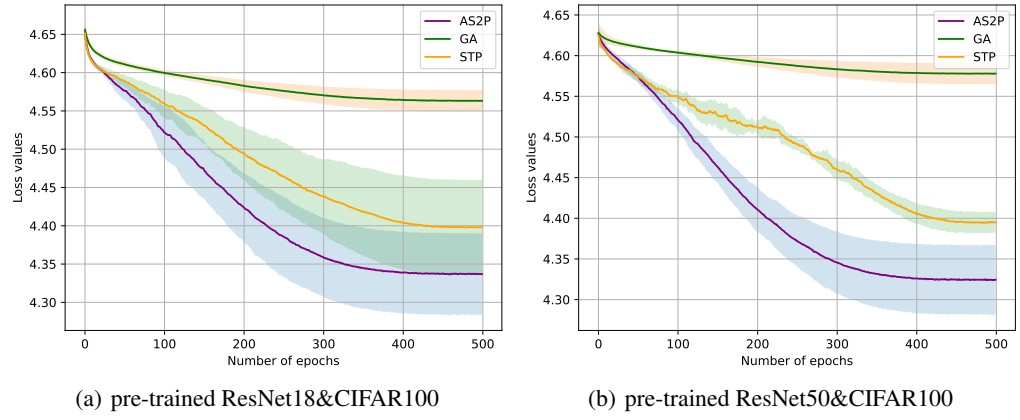

(a) pre-trained ResNet18&CIFAR100      (b) pre-trained ResNet50&CIFAR100

Figure 6: Verification of effectiveness of proposed method under pre-trained ResNet101&CIFAR100. Left-side figure demonstrated the convergence rate of AS2P without (W.O.) automatic learning rate and without progressive $\gamma$-clipping. Right-side two figures demonstrate the dynamics of learning rate and $\gamma$;

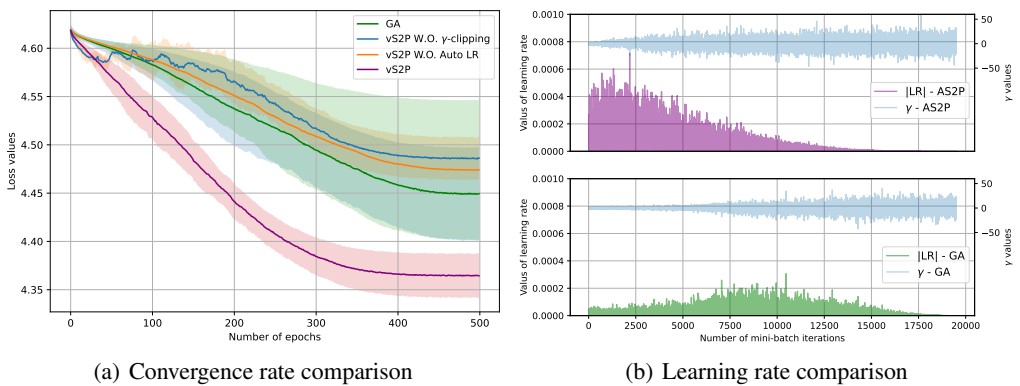

(a) Convergence rate comparison      (b) Learning rate comparison

Figure 7: Under ResNet18 and CIFAR10, the performance of GA with the different number of symmetric perturbations for each update. The left-side figure shows performance under the varying number of symmetric random perturbation per update where the number of function query for each setting are the same. The right-side figure demonstrates that under varying training settings, the convergence of GA with 10 symmetric random perturbations for gradient approximation per update. Basically, we can conclude that one symmetric symmetric random perturbation per update converges to smaller loss values under the same number of function queries.

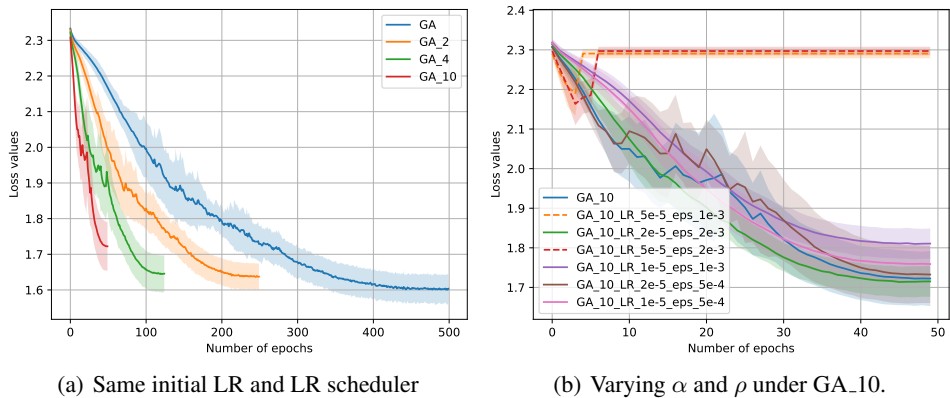

(a) Same initial LR and LR scheduler      (b) Varying $\alpha$ and $\rho$ under GA_10.

Figure 8: Performance comparison under VGG11 and CIFAR10. The left-side figure demonstrates the dynamics of training loss; The right-side figure demonstrates the training cost ratio of reaching the same specific loss values. The proposed method AS2P converges faster than other baseline methods and nearly requires $0.5\times$ number of queries to reach the same specific loss values. Note that the hyper-parameters directly follow the setting of ReSNet18&CIFAR10 in Table 3.

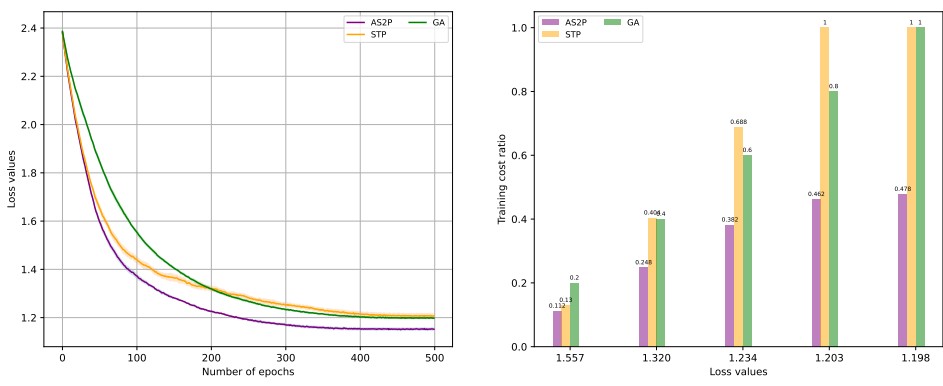

Figure 9: Performance comparison with various baselines under common deep models&datasets where the $x$-axis is the number of function queries. This figure is adopted from Figure 1.

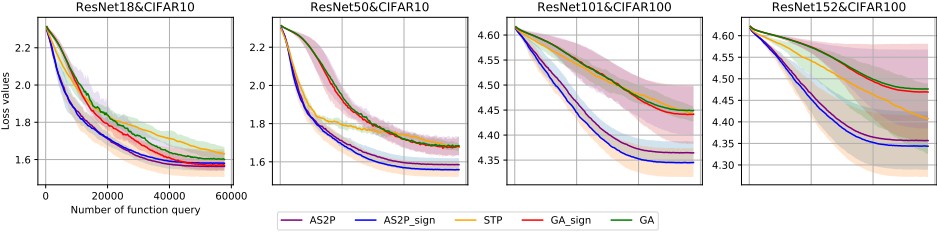

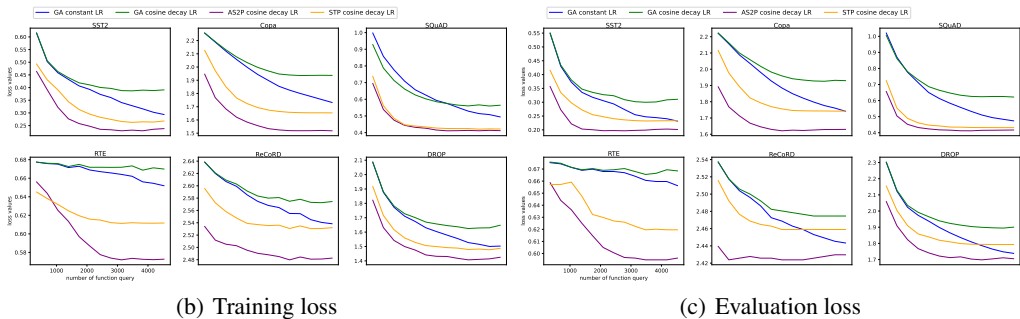

(b) Training loss                    (c) Evaluation loss

Figure 10: Performance comparison with full fine-tuning OPT-13B model where the $x$-axis is the number of function queries. This figure is adopted from Figure 2.

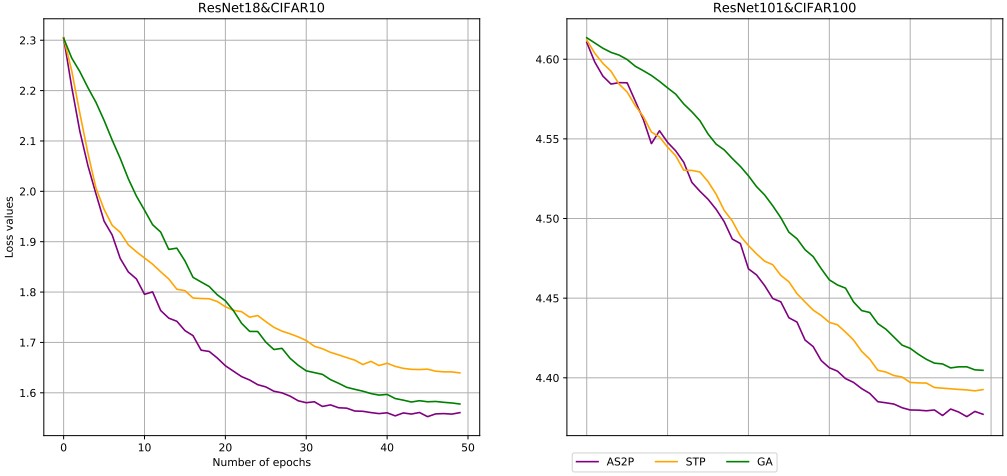

Figure 11: Corresponding validation performance of Figure 1(b) under setting ResNet18&CIFAR10 and ResNet101&CIFAR100. Using one seed only.

