# OpenReview forum: "Stochastic two points method for deep model gradient free optimization"
_ICLR.cc/2024/Conference — Submitted to ICLR 2024_

### Official Review · Reviewer_7rkr · 2023-10-29

**Soundness:** 2 fair
**Presentation:** 3 good
**Contribution:** 2 fair
**Rating:** 5
**Confidence:** 3

**Summary:**

In this submission, a zero-order optimiser, S2p, is proposed for efficient training for LLM along with theoretical analysis. Based on its theoretical findings, an accelerated version of S2P, AS2P, is studied by automatically tuning with learning rate while it is supported by two methods progressive \gama-cliplling and Automatic learning rate. The algorithm, AS2P, is tested on a variety of tasks including LLM training and some vision classification tasks with achieving noticeable improving margin for both training loss and convergence speed compared with the existing zero-order optimisers, STP and GA.

**Strengths:**

The proposed algorithm AS2P is compared with other zero-order-based optimisers on a variety of tasks showing a noticeable margin.
The paper is well-written and easy to understand.

**Weaknesses:**

1. The theoretical results are not directly related to the good performance of AS2P.
2. The recent work for gradient-free optimiser is not compared in the submission. For example,
Lin T, Zheng Z, Jordan M. Gradient-free methods for deterministic and stochastic nonsmooth nonconvex optimization. Advances in Neural Information Processing Systems. 2022 Dec 6;35:26160-75.

**Questions:**

1. Two main practical methods, progressive \gama-cliplling and Automatic learning rate, are introduced to increase the convergence rate which is also justified by the later experiments. It is not theoretically clear why AS2P has a better convergence rate than S2P. Does it mean these two methods are derived from the based on the previous theorem?
2. The imperial results show that the proposed algorithm especially AS2P has a better convergence rate compared with other zero-order algorithms, still first-order optimisers are commonly used in the practice, Do the authors base line for that?
3. From theorem 3.1 and theorem 3.2, four options are provided for setting \alpha_k iteratively. So what are the differences in practice? Is AS2P a special case of Option 4?
4. In all the experiments, AS2P outperforms GA and STP in terms of both the convergence speed and training and evaluation loss. Does the author give a fair hyperparameter tuning to all the competitors? And how the others are tuned? In the proposed theorem, there is no clear theoretical guarantee showing the convergence rate f(x)-f*(x) is improved in the paper can the author give more explanation?
5. When the zero-order optimiser is needed for LLM training? Without using gradient information how worse the trained model will be compared with that trained gradient-based optimiser such as AdamW?

Typo: In the last paragraph on Page 8. Figure 7(a) should be Figure 3(a)

---

> ### Author Response · Authors · 2023-11-18
>
> We greatly appreciate your constructive comments. Please see our responses below;
>
> $\textit{Weakness 1 and question 1,}$
>
> We justified our algorithm in Section 3.4, Especially, our work provides the first theoretical guarantee of zeroth-order optimization for deterministic (L0, L1)-smooth nonconvex optimization. Please refer to the first point of our official comments. Our theoretical results serve as the source of inspiration for our algorithm development, and the resultant algorithm, informed by these findings, is anticipated to better capture the dynamics of optimizing deep models compared to previous algorithms based on L-smoothness assumptions. Empirically, the improvements in our algorithm manifest in significant convergence enhancements, achieving the same loss values with nearly half the training costs.
>
>
> $\textit{For the weakness 2,}$
>
> The GA method in our experiment can be seen as the method in the referenced paper [1]. As far as our understanding goes, [1] is the first work to provide theoretical guarantees of the zeroth-order method for deterministic and stochastic nonsmooth nonconvex optimization under the (δ,ϵ)-Goldstein stationary optimality condition [2].
> Notably, [1] gives the positive result that the zeroth-order method can be used in solving nonsmooth nonconvex optimization but doesn't propose a practical algorithm for performance improvement. The algorithm used in [1], SGFM, is akin to basic zeroth-order optimization algorithms such as those in earlier works [3].
>
>
> $\textit{Question 2 and Question 5,}$
>
> There is no doubt that first-order methods are more commonly used in deep model problems. However, comparing first-order methods directly with zeroth-order optimization algorithms may not be suitable. Please refer to the second point of our official comments.
>
>
> $\textit{Question 3,}$
>
> Options 1 ~ 4 represent the foundational algorithms for our theoretical guarantee analysis.
> Take option 1 and option 2 as an example to illustrate the differences, which assume the L-smoothness and require the knowledge of $L$ for option 2, however, the property of the function $L$ is unknown in practice.
> So, if we choose Option 2 due to its theoretical improvement over Option 1 to optimize our objectives, then we need to tune the learning rate to find the best practical learning rate. The situation is similar to the situation of commonly used SGD, which also suggests knowledge of $L$ and learning rate scheduler $1/\sqrt{k}$ but we basically need to tune the learning rate for each task accordingly and commonly choose the learning rate scheduler as cosine decay.
> So it is true that the theoretical results, while not always directly applicable to real-world conditions, inform practical algorithm development. Notably, such as Option 4 and our proposed AS2P, theatrical findings in Option 4 are expected to improve practical performance by incorporating them into our algorithm developments.
>
>
> $\textit{Question 4,}$
>
> We highly appreciate your attention to ensuring a fair comparison of our experiment design. Due to space limitations, detailed experiment setup information is provided in Appendix D.1, as referenced in Section 4. Our effort to maintain fairness includes grid searches for hyperparameters (learning rate α and smoothing parameter ρ) for all baselines.
> Regarding f(x)-f*(x), for the smooth and non-convex optimization problems, the standard optimality condition is typically ϵ-first-order stationary point ( ||∇f(x)|| ≤ ϵ) given it is intractable to find global minima f*(x). We believe our improvements stem from deriving the algorithm under the relaxed smoothness assumption, which more closely captures deep model problems.
>
> Thank you for pointing out the typo (Figure 3a); it has been corrected. Your attention to detail is invaluable.
>
>
> [1] Lin T, Zheng Z, Jordan M. Gradient-free methods for deterministic and stochastic nonsmooth nonconvex optimization. Advances in Neural Information Processing Systems. 2022 Dec 6;35:26160-75.
>
> [2] Zhang, Jingzhao, et al. "Complexity of finding stationary points of nonconvex nonsmooth functions." International Conference on Machine Learning. PMLR, 2020.
>
> [3] Yurii Nesterov and Vladimir Spokoiny. Random gradient-free minimization of convex functions. Foundations of Computational Mathematics, 17:527–566, 2017.

---

> > ### Author Response · Authors · 2023-11-22
> >
> > Dear Reviewer 7rkr,
> >
> > We hope our response has answered your questions and clarified the concerns. If there are any further questions, please feel free to inform us before the closure of the interactive rebuttal system.
> >
> > Appreciate your time and happy Thanksgiving.

---

### Official Review · Reviewer_8MoD · 2023-11-01

**Soundness:** 3 good
**Presentation:** 3 good
**Contribution:** 2 fair
**Rating:** 5
**Confidence:** 3

**Summary:**

The paper introduces a new method called Stochastic Two-Point (S2P) for gradient-free optimization of large deep models, including language models. The authors analyze the convergence properties of S2P under general and relaxed smoothness assumptions and propose a faster variant called Accelerated S2P (AS2P). Extensive experiments demonstrate that AS2P outperforms standard methods and achieves significant speed-up in training large deep models. The contributions of the paper include novel insights into query complexity and the development of an efficient optimization approach for models with limited computational resources.

**Strengths:**

1. This paper has provided strong theoretical support for the newly proposed algorithms S2P and AS2P.
2. This paper is generally well presented.

**Weaknesses:**

1. This paper does not give a convincing motivation of using L0,L1-smoothness for the proof of zeroth-order optimization algorithms, which therefore make it hard to know whether it is really necessary to give another proof (compared with using the existing ones).
2. The comparison and discussion on other existing zeroth-order optimization algorithms of this paper are limited, e.g., GLD[R1], ZoRD[R2].
3. The performance of the S2P and AS2P seems to be limited as they are only able to reduce the training loss slightly, which is significantly worse than the first order optimization as shown in Fig.1b.
4. While this paper is motivated by LLM training, this paper does not give any empirical experiments to show that it indeed can help solve the problem in LLM training.

[R1] Golovin, D., Karro, J., Kochanski, G., Lee, C., Song, X., & Zhang, Q. (2019). Gradientless descent: High-dimensional zeroth-order optimization. arXiv preprint arXiv:1911.06317.

[R2] Shu, Y., Dai, Z., Sng, W., Verma, A., Jaillet, P., & Low, B. K. H. (2022, September). Zeroth-Order Optimization with Trajectory-Informed Derivative Estimation. In The Eleventh International Conference on Learning Representations.

**Questions:**

1. Why the L0,L1-smoothness will be more realistic in practice especially for the training of LLM? Any empirical or theoretical support?
2. Why the convergence proof of this paper requires that 3.2 and 3.3 are satisfied at the same time? From my understanding 3.2 can infer that L0=L and L1=0 in 3.3.
3. Can the authors provide a figure that compares the convergence w.r.t. number of queries during optimization among S2P/AS2P and other ZOO algorithms.

---

> ### Author Response · Authors · 2023-11-18
>
> $\textit{Weakness 1 and question 1,}$
>
> We appreciate your acknowledgment of our theoretical guarantee of zeroth-order optimization for deterministic (L0, L1)-smooth nonconvex optimization. Please refer to the first point of our official comments.
> Some existing works do provide empirical evidence to show the previous L-smoothness is often not satisfied in practical machine-learning problems [1, 2]. Notably, our work aligns with this trend and emphasizes the precision of (L0, L1)-smoothness in characterizing objective function landscapes, especially in training deep neural networks.
>
> $\textit{Weakness 2,}$
>
> The works cited ([R1, R2]) differ significantly from ours in scope. Their experiments typically involve small problem dimensions, such as d = 32*32 for black-box attacks on CIFAR10 and a MLP with d = 2189 for non-differentiable metric optimization. In contrast, our experiments optimize ResNet (d = 11.7M) and even LLM (d = 13B). Implementing the baseline method Algorithm 1 in [R2] for our large-scale problems would be impractical, we have some comparisons below. (Note for simplification we use the baseline Algorithm 1 in [R2] instead of the proposed method in [R2] since they have similar performance.)
>
> Under ResNet18, CIFAR10, batch size 1024, and same lab hardware,
>
> AS2P completing 500 epoch training requires ~1h.
>
> Algorithm 1 in [R2] taking one step of parameter updating requires ~ 195h (Finishing 1 epochs training, then ~476 days)
>
> A quick answer to this large difference is that our method uses two time function queries to approximate the gradient while algorithm 1 in [R2] uses dimension d time function queries to approximate the gradient. However, the effectiveness of approximating gradients with two-symmetric perturbations is an ongoing research topic [3,4].
>
> $\textit{Weakness 3 and 4,}$
>
> Comparing first-order methods directly with zeroth-order optimization algorithms may not be suitable. Please refer to the second point of our official comments.
> Additionally, we adopted the hardware budget for different methods from [3] to demonstrate the effectiveness of applying the zeroth-order method in LLM training, for instance, a A100 can fully fine-tune the largest OPT model with 2.7B parameters, FT-prefix up to 6.7B, and zeroth-order one can go up to fine-tune 30B model (Table 4 in [3], adopted in Section 4 of our work).
>
>
> $\textit{Question 2,}$
>
> Apologies for any confusion. We introduced Definition 3.2 and Definition 3.3 at the beginning, not implying that both are satisfied simultaneously. Specifically, assumption 1 (Definition 3.2), is a popular one for deep model convergence analysis, and Theorem 3.1 satisfies assumption 1 only. Assumption 2 (Definition 3.3) is an emerging research topic, and Theorem 3.2 satisfies assumption 2 only. Those two assumptions lead to two distinct algorithms, mainly distinguished by the linear and non-linear correlations between step size α and |γ|. Those relationships inspire our practical algorithm developments.
>
> $\textit{Question 3,}$
>
> We appreciate your careful examination in terms of our experiments and propose an important and concerning question for most readers we believe.
> To address concerns about practical query complexity, we designed Figure 1a and Figure 2a specifically to illustrate performance under the same number of queries (or training cost ratio, as mentioned in the figures). Additionally, we added the figures demonstrating the convergence w.r.t. number of queries during optimization as Figure 9 and Figure 10 in Appendix D.2. (See the uploaded revised PDF).
>
> We greatly appreciate your constructive comments and are open to addressing any further concerns or questions you may have.
>
>
> [1] Faw, Matthew, et al. "Beyond uniform smoothness: A stopped analysis of adaptive sgd." The Thirty Sixth Annual Conference on Learning Theory. PMLR, 2023.
>
> [2] Zhang, Bohang, et al. "Improved analysis of clipping algorithms for non-convex optimization." Advances in Neural Information Processing Systems 33 (2020): 15511-15521.
>
> [3] Malladi, Sadhika, et al. "Fine-Tuning Language Models with Just Forward Passes." arXiv preprint arXiv:2305.17333 (2023).
>
> [4] Yue, Pengyun, et al. "Zeroth-order Optimization with Weak Dimension Dependency." The Thirty Sixth Annual Conference on Learning Theory. PMLR, 2023.

---

> > ### Author Response · Authors · 2023-11-22
> >
> > Dear Reviewer 8MoD,
> >
> > We hope our response has answered your questions and clarified the concerns. If there are any further questions, please feel free to inform us before the closure of the interactive rebuttal system.
> >
> > Appreciate your time and happy Thanksgiving.

---

### Official Review · Reviewer_QZx8 · 2023-11-03

**Soundness:** 3 good
**Presentation:** 2 fair
**Contribution:** 2 fair
**Rating:** 6
**Confidence:** 3

**Summary:**

The authors are striving to propose a gradient-free learning method, called Stochastic Two-Point. The method is derived from the Stochastic Three-Point approach, where the proposed method ignores the current weight within the three-point set during optimization. The authors give some theoretical analysis in regards to the convergence of the proposed two-point method. And to show the effectiveness of the proposed method, the authors give results on Cifar dataset and some nlp tasks.

**Strengths:**

**Strengths**

1. The paper is clearly written and easy to follow.
2. It is interesting to see a forward-only method for optimization. Although currently such methods may give much worse results compared to bp, I think they are very promising in tackling problems in specific scenarios, such as optimizing non-differentiable loss.
3. From the experiment results, the proposed method could give comparable results.

**Weaknesses:**

**Weakness**

1. This paper basically follows the paper of three point method. The key difference is that this paper ignores the current weight when solving the argmin in optimization. But in the STP method, it is crucial that the current weight gives guarantee on the monotonicity of optimization. Without such guarantee, it seems quite not likely that the method could reach a suboptimal condition, i.e. $min ||g|| \leq \epsilon$. Given that the current weight is ignored in the proposed two point method, how the authors ensure the minimization is non-increasing? I have not found any discussions in regards to the failure cases.

    I notice that the Theorem 3.1 presents a similar conclusion in regards to that in STP. The theorem actually implies that the method could reach at suboptimal under certain condition. But to my understanding, the convergence properties could be worse than the STP method, where the proposed method is proved to converge over estimation. Could the authors give detailed comparisons between the proposed method and STP, given that the two methods are quite similar.

2. I am wondering could the authors provide some evidence other than the results that the proposed method could give better approximation compared to other similar methods? That would be more substantial to support the superiority of your methods. For example, the authors could measure the distance between the optimal model and the model trained with the proposed method.

3. Regarding the experiments, firstly the authors may need to compare some other advanced similar methods, like the MEZO (Sadhika Malladi) cited by the authors. Secondly, the evaluation performance is missing regarding training on the Cifar datasets. Thirdly, I think the authors may need to add results of using bp gradient to give reference. Fourthly, the figures are quite not friendly to read, to present clear comparison, the authors may provide tables in addition. Finally, it is highly recommended that the authors can release their code for reproductivity.

**Questions:**

See weakness

**Details Of Ethics Concerns:**

I have not found any discussions about the limitations and potential negative societal impact. But in my opinion, this may not be a problem, since the work only focuses on the learning method. Still, it is highly encouraged to add corresponding discussions.

---

> ### Author Response · Authors · 2023-11-18
>
> Thank you for taking the time to read and review our paper. We sincerely appreciate your keen interest in the technical aspects of our work. Please see our responses below;
>
> Weakness 1,
>
>
> I believe there might be a misunderstanding regarding how stochastic optimization algorithms achieve optimality conditions. Taking results in two recent works as examples, such as Theorem 5 in [1] (zeroth-order optimization) and Theorem 2 in [2] (stochastic first-order optimization, mini-batch), the optimality conditions are achieved in expectation or with certain probabilities. It is well-established that having progress at each step for stochastic optimization algorithms is neither guaranteed nor necessary. Progressive monotonicity during optimization for stochastic optimization algorithms is rarely a focus, as seen in the golden baseline [3] referencing the GA method in our work.
> Therefore, the perceived "overestimation" is not a bottleneck for our algorithm. In comparison with the STP algorithm, we eliminate the non-updating component of STP, effectively saving one forward pass in a batch data forward pass. The practical performance of our proposed method surpasses standard methods with a 2× speed-up in training across most conducted tasks. Additionally, please refer to the first point of our official comments.
>
> Weakness 2,
>
> The mentioned distance $||x_{k} - x_{0}||$ is meaningful in convex problems and usually appears in the corresponding convergence analysis. To the best of our knowledge, there is no theoretical evidence to show the effectiveness of the distance in smooth non-convex problems.
>
> Weakness 3,
>
> 3.1 As outlined in Section 4, it is important to note that 'GA is principally equivalent to MeZO as presented in Malladi et al. (2023), which reduces memory consumption with an implementation trick by performing twice forward passes sequentially instead of in parallel.' Hence, the GA method in our experiment aligns exactly with the MeZO method.
>
> 3.3 Comparing first-order methods directly with zeroth-order optimization algorithms may not be suitable. Please refer to the second point of our official comments.
>
> 3.2, 3.4, and 3.5 We have started working on the corresponding changes such as the additional table version result (Table 5 and Table 6 in Appendix D.2) and evaluation loss regarding the cifar datasets (Figure 1  in Appendix D.2). Please see the uploaded revised PDF. If you have any further concerns, we would like to address them as well. And, the code will be released upon acceptance.
>
>
>
> [1] Lucchi, Aurelien, Antonio Orvieto, and Adamos Solomou. "On the second-order convergence properties of random search methods." Advances in Neural Information Processing Systems 34 (2021): 25633-25645.
>
> [2] Wang, Bohan, et al. "Convergence of adagrad for non-convex objectives: Simple proofs and relaxed assumptions." The Thirty Sixth Annual Conference on Learning Theory. PMLR, 2023.
>
> [3] Yurii Nesterov and Vladimir Spokoiny. Random gradient-free minimization of convex functions. Foundations of Computational Mathematics, 17:527–566, 2017.

---

> > ### Comment · Reviewer_QZx8 · 2023-11-20
> > **Thanks for the response.**
> >
> > I have read the response, and thanks for the effort. It is clear that one will save computation budget if you choose to ignore one forward propagation per batch. However, given the poor results gradient-free may give, I think it would be more concerned to give investigation on how to improve the performance of such algorithms. In this paper, the authors even choose to not report the testing performance of Cifar. In my opinion, it can be much more meaningful to study how to reduce such huge gap with gradient descent. Overall, I think the paper has not focusing on the core issues in the corresponding domain, thus it has not made a significant contribution to advancing this domain, especially considering the authors have not presented interesting pratical results. Thanks for the explanation, and based on the previous points, I raise my rating to 6.

---

### Official Review · Reviewer_v68Q · 2023-11-03

**Soundness:** 2 fair
**Presentation:** 3 good
**Contribution:** 2 fair
**Rating:** 5
**Confidence:** 2

**Summary:**

In this work, the authors propose a stochastic two-point method for optimizing deep neural networks without resorting to gradient descent optimization techniques. Besides, the same authors propose an even accelerated version of it and compare it to other gradient-free approaches. The authors propose convergence analysis and conduct some experiments.

**Strengths:**

- the work proposes a new gradient-free optimization method. In a world computationally-constrained, this is prospectively a promising research path
- although I could not check the correctness, the authors make a big effort to ground the approach
- the authors even tested on a LLM

**Weaknesses:**

- no comparison (apparently) with standard gradient descent approaches
- empirical analysis limited to relatively small tasks using relatively large architectures
- (minor) graphics is barely readable, and so the results

**Questions:**

- how is the proposed approach performing, compared with traditional gradient descent (with for example SGD or Adam)?
- how is the approach performing in more dataset-challenging scenarios where the model is not able to overfit on the given task (like ResNet-18 trained on ImageNet1k)?
- how is the approach performing with architectures without skip connections?
- how much extra memory is the proposed approach consuming compared to other competitors?

---

> ### Author Response · Authors · 2023-11-18
>
> Thank you for the constructive comments. The following addresses the weaknesses and questions.
>
> We have worked on the corresponding changes, such as the additional table version result to “zoom in” the graphics (See Table 5 and Table 6 of Appendix D.2 in the revised PDF). If you have any further concerns, we would like to address them as well.
>
> $\textit{Weakness 1 and Question 1}$
>
> The first-order methods and zeroth-order optimization algorithms have different use scenarios and may not be directly comparable.
> Please refer to the second point of our official comments below.
> $\textit{Performance comparison with first-order method}$
> Firstly, we conduct a simple experiment with the first-order method (Adam) under ResNet18 and CIFAR10, similar to Figure 1b.
>
> Adam decreases the training loss from ~2.3 to ~0.0 in $\textbf{10}$ epochs.
>
> AS2P decreases the training loss from ~2.3 to ~1.6 in $\textbf{500}$ epochs.
>
> However, this huge gap is not evidence of negating zeroth-order optimization since zeroth-order optimization algorithms have distinct applications, especially in optimizing non-differentiable objectives and scenarios with constrained hardware budgets. In the context of LLM applications, our method aligns with this trend, exemplified by [1], showcasing up to 12× memory reduction compared to first-order methods. Our experiments, following the LLM training settings in [1], demonstrate nearly a 2× speed-up over baseline methods under the same hardware constraints, reinforcing the efficacy of zeroth-order optimization for fully fine-tuning LLMs within inference hardware budgets.
>
>
>
> $\textit{Weakness 2, Question 2, and Question 3,}$
>
> We appreciate the suggestions regarding our experiment setup. The experiments in our original submission have encompassed a range of common deep models and datasets, including ResNet & CIFAR, and LLMs, which involve popular ResNet structure and attention structure. Across various settings and tasks, our proposed method consistently outperforms other baselines by a substantial margin.
>
> Following your suggestion, we conducted additional experiments with VGG (architectures without skip connections), and the results show that the proposed algorithm consistently outperforms baselines by a large margin and requires 0.5$\times$ number of queries to reach specific loss values, which is summarized in Figure 8 of Appendix D.2 (See uploaded revised PDF). Though we do not have enough time to test ImageNet in this period, we believe our experiments covered common conditions and provided robust evidence of the method's advantages.
>
>
> $\textit{Question 4,}$
>
> In our experiments, all mentioned methods consume the same amount of memory, ensuring consistency in comparison.
>
>
> [1] Malladi, Sadhika, et al. "Fine-Tuning Language Models with Just Forward Passes." NeurIPS 2023.

---

> > ### Comment · Reviewer_v68Q · 2023-11-22
> > **Thank you for your answer**
> >
> > Dear author(s),
> > Many thanks for providing a response to the initial comments. In particular, the answer to W2 is satisfactory.
> > Regarding W1, it is unclear whether the proposed approach gives a speedup compared to traditional GD approaches - in your example, Adam is able to minimize the loss to zero in 10 epochs, while AS2P fails in that despite more than a magnitude order more epochs. Evidently, the proposed approach is better suited for memory-limited applications (at the cost of big performance loss); however, this opens the door to comparison with other strategies that employ GD but consume less memory/computation (like training a subgraph only or employing adapters/prompts... still this method seems not to be ready to deployment in real life, compared to other traditional approaches. For this reason, I confirm my initial score.

---

> > > ### Author Response · Authors · 2023-11-22
> > >
> > > We thank the reviewer for the follow-up comments and bringing parameter-efficient tuning to our attention.
> > > It is indeed that the proposed method is better suited for memory-limited application scenarios. Our method enables $\textbf{fully fine-tuning}$ LLMs within inference hardware budgets, and we believe it can be embedded and mutually enhanced with traditional parameters-efficient approaches.
> > >
> > > Please note that the main contributions of our work are the theoretical guarantee for (L0, L1)-smooth nonconvex problems, practical training accelerations compared with other zeroth-order methods, and providing $\textbf{extra}$ tools on LLM fine-tunings.
> > >
> > > We hope this can address your concern regarding the usage of our work. We are glad to have further discussion on this point.

---

### Author Response · Authors · 2023-11-18

Thank all reviewers for taking the time to read and review our paper. We’d like to highlight the following two points.

To the best of our knowledge, our study is pioneering in presenting a theoretical guarantee of zeroth-order optimization for deterministic (L0, L1)-smooth nonconvex optimization. The (L0, L1)-smoothness assumption is gaining wide acceptance as it more closely captures deep model problems than the L-smoothness and is progressively becoming a standard consideration in the convergence analysis of deep models [1,2,3,4].

$\textit{Performance comparison with first-order method}$
Firstly, we conduct a simple experiment with the first-order method (Adam) under ResNet18 and CIFAR10, similar to Figure 1b.

Adam decreases the training loss from ~2.3 to ~0.0 in $\textbf{10}$ epochs.

AS2P decreases the training loss from ~2.3 to ~1.6 in $\textbf{500}$ epochs.

However, this huge gap is not evidence of negating zeroth-order optimization since zeroth-order optimization algorithms have distinct applications, especially in optimizing non-differentiable objectives and scenarios with constrained hardware budgets. In the context of LLM applications, our method aligns with this trend, exemplified by [5], showcasing up to 12× memory reduction compared to first-order methods. Our experiments, following the LLM training settings in [5], demonstrate nearly a 2× speed-up over baseline methods under the same hardware constraints, reinforcing the efficacy of zeroth-order optimization for fully fine-tuning LLMs within inference hardware budgets.




[1] Faw, Matthew, et al. "Beyond uniform smoothness: A stopped analysis of adaptive sgd." The Thirty Sixth Annual Conference on Learning Theory. PMLR, 2023.

[2] Zhang, Bohang, et al. "Improved analysis of clipping algorithms for non-convex optimization." Advances in Neural Information Processing Systems 33 (2020): 15511-15521.

[3] Wang, Bohan, et al. "Convergence of adagrad for non-convex objectives: Simple proofs and relaxed assumptions." The Thirty Sixth Annual Conference on Learning Theory. PMLR, 2023.

[4] Chen, Ziyi, et al. "Generalized-Smooth Nonconvex Optimization is As Efficient As Smooth Nonconvex Optimization." ICML 2023

[5] Malladi, Sadhika, et al. "Fine-Tuning Language Models with Just Forward Passes." NeurIPS 2023.

---

### Meta-Review · Area_Chair_6KYB · 2023-12-04

**Metareview:**

The paper proposes a stochastic two-point method for optimizing deep neural networks. This new variant that which eliminates the non-updating component $f(x_k)$ of an earlier method called STP (and is therefore computationally cheaper). The paper also proposes an accelerated variant. The paper mostly claims two contributions: 1) new theoretical convergence guarantees for $(L_0, L_1)$ smooth functions (which include a broader class of functions compared to $L$-smooth functions), and 2) practical training accelerations compared with other zeroth-order methods.

Regarding 1), one aspect that is not well discussed is whether one obtains any significant benefit from $(L_0, L_1)$ smoothness. To the best of my knowledge, such functions were first introduced in https://arxiv.org/pdf/1905.11881.pdf to demonstrate that clipping methods can be faster than methods that don't use clipping. However, I don't see any similar argument here. Another aspect to consider is that the paper does not seem to have any theoretical guarantee for the accelerated version of their algorithm.

Regarding 2), several reviewers think the paper lacks motivation for training LLMs. Importantly,  [1] actually made a similar contribution to speed up derivative-free methods so the paper seems to slightly oversell their contribution from an empirical point of view. However, the empirical results do show some significant gains over the method called GA, which is claimed to be similar to Mezo (I think it would be less confusing to have an actual implementation of Mezo)

Overall, I agree with the reviewers that the theoretical contribution of the paper lacks motivation. What are the benefits of the $(L_0, L_1)$ smoothness assumption on the rate of convergence? A revision of the paper needs to provide an extended comparison and discussion of the theorems derived from the paper.

Although I'm not able to recommend acceptance at this stage, I think the paper does pursue an interesting direction but it needs more work before publication.

[1] Malladi, Sadhika, et al. "Fine-Tuning Language Models with Just Forward Passes." NeurIPS 2023.

**Justification For Why Not Higher Score:**

Overall, the paper is not clear on what are the main contributions. It needs some significant rewriting.

**Justification For Why Not Lower Score:**

N/A

---

### Decision · Program_Chairs · 2024-01-16

Reject